



# Wind-driven Emission of Marine Ice Nucleating Particles in the Scripps Ocean-Atmosphere Research Simulator (SOARS)

Kathryn A. Moore[1,*], Thomas C. J. Hill[1], Samantha Greeney[2,^], Chamika K. Madawala[3], Raymond J. Leibensperger III[4], Christopher D. Cappa[5], M. Dale Stokes[4], Grant B. Deane[4], Christopher Lee[4], Alexei V. Tivanski[3], Kimberly A. Prather[4,6], and Paul J. DeMott[1]

[1]Department of Atmospheric Science, Colorado State University, Fort Collins, CO, USA
[2]Department of Atmospheric Sciences, Texas A&M University, College Station, TX, USA
[3]Department of Chemistry, University of Iowa, Iowa City, IA, USA
[4]Scripps Institution of Oceanography, University of California San Diego, La Jolla, CA, USA
[5]Department of Civil and Environmental Engineering, University of California Davis, Davis, CA, USA
[6]Department of Chemistry and Biochemistry, University of California San Diego, La Jolla, CA, USA
[*]Now at: Earth System Science Interdisciplinary Center, University of Maryland, College Park, and NASA Goddard Space Flight Center, Greenbelt, MD, USA
[^]Now at: Department of Atmospheric Science, Colorado State University, Fort Collins, CO, USA

**Correspondence:** Kathryn A. Moore (kathryn.a.moore@colostate.edu)

**Abstract.** Sea spray aerosol (SSA) represent one of the most abundant natural aerosol types, contributing significantly to global aerosol mass and aerosol optical depth, as well as to both the magnitude and uncertainty of aerosol radiative forcing. In addition to their direct effects, SSA can also serve as ice nucleating particles (INPs), which are required for the initiation of cloud glaciation at temperatures warmer than ∼-36 °C. This study presents initial results from the CHaracterizing Atmosphere-Ocean

parameters in SOARS (CHAOS) mesocosm campaign, which was conducted in the new Scripps Ocean-Atmosphere Research Simulator (SOARS) wind-wave channel at the Scripps Institution of Oceanography. SOARS allows for isolation of individual factors, such as wave height, wind speed, water temperature, or biological state, and can carefully vary them in a controlled manner. Here, we focus on the influence of wind speed on the emission of SSA and INPs. Unlike recent measurements from the Southern Ocean, real-time and offline INP observations during CHAOS exhibited opposite relationships with wind speed,

which may be related to sampling inlet differences. Changes in the INP activated fraction, dominant INP particle morphology, and INP composition were seen to vary with wind. Seawater ice nucleating entity concentrations during CHAOS were stable over time, indicating changes in atmospheric INPs were driven by wind speed and wave-breaking mechanics rather than variations in seawater chemistry or biology. While specific emission mechanisms remain elusive, these observations may help explain some of the variability in INP concentration and composition that have been seen in ambient measurements.

## 1 Introduction

Sea spray aerosol (SSA) are marine-derived particles composed of mixtures of inorganic salts and organic compounds, with the exact composition and mixing state varying based on particle size, production mechanism, and the underlying biology and geochemistry of the source seawater (e.g. Lewis and Schwartz, 2004; O'Dowd and de Leeuw, 2007; de Leeuw et al.,



2011; Cochran et al., 2017). Along with mineral and soil dusts, SSA dominates atmospheric aerosol mass, and contributes
∼30 % to globally-averaged total aerosol optical depth (AOD) (O'Dowd and de Leeuw, 2007; Bellouin et al., 2013). SSA are
generated through wind stress at the ocean surface, either through the direct tearing of breaking wave crests (spume drops)
or as a result of bubble bursting (film and jet drops) following air entrainment during wave breaking (Lewis and Schwartz,
2004; O'Dowd and de Leeuw, 2007; Deike et al., 2022). Additionally, oxidation of dimethyl sulfide (DMS) and other biogenic
volatile organic compounds (BVOCs) emitted from the ocean can lead to the condensation of gas-phase species and formation
of secondary marine aerosol (SMA) (Lewis and Schwartz, 2004; Quinn et al., 2017; Naik et al., 2021). Given their ubiquity in
the atmosphere, SSA are an important contributor to both the magnitude and uncertainty of aerosol radiative forcing (Andreae,
2007; Carslaw et al., 2013, 2017; Forster et al., 2021).

The indirect radiative impact of both SSA and SMA through their role as cloud condensation nuclei (CCN) has received
considerable attention from observational, laboratory, and modeling studies (e.g. Pierce and Adams, 2006; Andreae, 2007;
Grythe et al., 2014; Modini et al., 2015; McCoy et al., 2015a; Quinn et al., 2017; Heinze et al., 2019; Mayer et al., 2020;
Gryspeerdt et al., 2023). Spurred by observations in remote ocean regions and laboratory mesocosm studies (Rosinski et al.,
1987; Bigg, 1973, 1990; Knopf et al., 2011; DeMott et al., 2016), the contribution of marine aerosol to the ice nucleating
particle (INP) budget, and thus indirectly to cloud phase, has come under increasing focus in recent years (e.g. Burrows et al.,
2013; Wilson et al., 2015; Irish et al., 2017; Vergara-Temprado et al., 2017, 2018; McCluskey et al., 2018c, b, a; Welti et al.,
2018; Huang et al., 2018; Creamean et al., 2019; McCluskey et al., 2019; Schmale et al., 2019; Irish et al., 2019; Welti et al.,
2020; Ickes et al., 2020; Hartmann et al., 2020, 2021; Zhao et al., 2021; Mitts et al., 2021; Tatzelt et al., 2022; Alpert et al.,
2022; Steinke et al., 2022; Raatikainen et al., 2022; Lin et al., 2022; McCluskey et al., 2023; Raman et al., 2023; Miyakawa
et al., 2023; Kawana et al., 2024). INPs are critical in initiating cloud glaciation at temperatures warmer than ∼-36 °C and
thus exert a large influence on cloud properties related to phase, such as lifetime, precipitation formation, and radiative forcing
(e.g. Kanji et al., 2017). Additionally, mixed-phase clouds, which contain both liquid and ice, play major roles in determining
cloud feedbacks (McCoy et al., 2015b, 2016), global cloud radiative properties (Cesana and Storelvmo, 2017), and equilibrium
climate sensitivity (Zelinka et al., 2020; Bjordal et al., 2020).

Measurements of ice nucleation in marine environments were first made in the late 1950s and 1960s (see Ickes et al., 2020,
Table 1). Since then, a few studies have suggested whole phytoplankton cells or marine bacteria may be the ice nucleating
components of SSA (Fall and Schnell, 1985; Knopf et al., 2011; Wilbourn et al., 2020; Beall et al., 2021). However, the
majority of studies indicate marine macromolecules, phytoplankton exudates, or other biogenic, organic species are the ice
nucleating components based on the generally small size (<0.2 μm) of ice nucleating entities in seawater, and their relationship
with biological activity (Schnell and Vali, 1976; Rosinski et al., 1987; Knopf et al., 2011; Wilson et al., 2015; Wang et al.,
2015; Ladino et al., 2016; DeMott et al., 2016; Irish et al., 2017; McCluskey et al., 2018b; Alpert et al., 2022; Hill et al.,
2023). Based on laboratory and mesocosm experiments, several studies have also inferred different components may be active
at different temperatures, as well as at different times throughout the onset and decay of phytoplankton blooms (DeMott et al.,
2016; McCluskey et al., 2018b; Ickes et al., 2020). In addition to the small and ubiquitous marine organic INPs, a second
category of more intermittent, larger, and heat sensitive marine INPs that are active at warmer temperatures has been identified



(McCluskey et al., 2018b; Ickes et al., 2020; Hartmann et al., 2020; van Pinxteren et al., 2020). These may be associated with
microbes or cellular debris, but have not been definitively identified. Recent laboratory studies have pointed to the importance
of supermicron SSA as a marine INP (Mitts et al., 2021), however, no assessment of the atmospheric transport of such particles
was conducted and ambient observations have yet to confirm this.

INP concentrations in remote marine regions are generally several orders of magnitude lower than those in continental areas
(DeMott et al., 2016; McCluskey et al., 2018c; Welti et al., 2020; Tatzelt et al., 2022). Based on normalization by particle num-
ber or surface area, marine INPs are also significantly less efficient at nucleating ice than species such as mineral or soil dusts
(DeMott et al., 2016; Kanji et al., 2017; McCluskey et al., 2018c). Despite this, in remote areas such as the Southern Ocean,
marine INPs are hypothesized to be the dominant contributor to the INP budget due to the lack of continental influence (Bur-
rows et al., 2013; Vergara-Temprado et al., 2017, 2018; McCluskey et al., 2019), and may dominate seasonally or intermittently
in other regions such as the high Arctic (Huang et al., 2018; Creamean et al., 2019; Hartmann et al., 2020; Ickes et al., 2020;
Hartmann et al., 2021). Atmospheric concentrations of the small, organic marine INP type were parameterized using observa-
tions from Mace Head in the North Atlantic (McCluskey et al., 2018c), and subsequent implementation in CAM5 (Community
Atmosphere Model version 5) and CAM6 (version 6) compared well to observations made in the Southern Ocean (McCluskey
et al., 2019, 2023). Other recent modeling work has focused on the intermittent, high temperature marine INPs (Steinke et al.,
2022), or freezing kinetics of background SSA particles (Alpert et al., 2022). Despite these efforts, the fundamental factors
controlling the emission of marine INPs from the sea surface to the atmosphere remain largely unknown.

Significantly more is known about the factors influencing the production of sea spray, although there is still huge variability
in simulated SSA fluxes among models, especially in polar regions (de Leeuw et al., 2011; Grythe et al., 2014; Deike et al.,
2022; Lapere et al., 2023). Numerous parameterizations for sea spray size distribution functions have been proposed (e.g.
Monahan and Muircheartaigh, 1980; Monahan et al., 1986; Gong, 2003; Mårtensson et al., 2003; Lewis and Schwartz, 2004;
de Leeuw et al., 2011; Sofiev et al., 2011; Jaeglé et al., 2011; Meskhidze et al., 2013; Ovadnevaite et al., 2014; Grythe et al.,
2014; Salter et al., 2015), with the choice influencing not only emitted SSA number and mass, but also the simulated radiative
budget and aerosol-cloud interactions once implemented in models (Grythe et al., 2014; McCoy et al., 2015a; Barthel et al.,
2019; Johnson et al., 2020). Although wind speed is the dominant influence on SSA production (Lewis and Schwartz, 2004;
O'Dowd and de Leeuw, 2007; de Leeuw et al., 2011), other factors including sea surface temperature (Mårtensson et al., 2003;
Sellegri et al., 2006; Jaeglé et al., 2011; Zábori et al., 2012; Ovadnevaite et al., 2014; Salter et al., 2014, 2015; Schwier et al.,
2017; Forestieri et al., 2018; Saliba et al., 2019; Barthel et al., 2019; Christiansen et al., 2019; Hartery et al., 2020; Liu et al.,
2021; Zinke et al., 2022; Sellegri et al., 2023), salinity (Mårtensson et al., 2003; Zábori et al., 2012; Ovadnevaite et al., 2014;
May et al., 2016; Nilsson et al., 2021; Zinke et al., 2022), and seawater biology and chemistry (O'Dowd et al., 2004; Sellegri
et al., 2006; Fuentes et al., 2010; Wang et al., 2015; McCoy et al., 2015a; Schwier et al., 2017; Burrows et al., 2018; Forestieri
et al., 2018; Saliba et al., 2019; Christiansen et al., 2019; Sellegri et al., 2023) have also been shown to influence production.
Conflicting and sometimes contradictory results for the magnitude and even sign of the impact of each of these variables has
been observed in laboratory and field measurements, which has not aided evaluation of the numerous available SSA source
parameterizations.




The new Scripps Ocean-Atmosphere Research Simulator (SOARS) wind-wave channel at the Scripps Institution of Oceanography, University of California San Diego was designed to tackle some of these outstanding questions about the production and atmospheric impacts of SSA. This study focuses on first results from the SOARS channel during the CHaracterizing Atmosphere-Ocean parameters in SOARS (CHAOS) mesocosm campaign, conducted for two months in 2022. The overarching goal of CHAOS was to understand and reduce uncertainty in the impact of wind speed on SSA production. Improvements over previous wave channel experiments (Prather et al., 2013; Wang et al., 2015; Sauer et al., 2022) include the ability to modulate wind speed in the wave channel, increasing atmospheric relevance. This study will touch on SSA production in SOARS, but primarily address the role of wind speed in emissions of marine INPs, which has not previously been characterized through controlled experiments.

## 2 Methods

### 2.1 Production of Sea Spray Aerosols in SOARS

Measurements described in this study were collected during the CHaracterizing Atmosphere-Ocean parameters in SOARS (CHAOS) study, during August 2022. SSA were produced in the new Scripps Ocean-Atmosphere Research Simulator (SOARS) wind-wave channel at the Scripps Institution of Oceanography (SIO), which is shown schematically in Fig. A1. The SOARS wave channel is 2.4 m wide, 2.4 m tall, and 36 m in length, with a nominal water volume of 103,680 L when filled. This is approximately 9 times the water volume of the glass wave channel described in Sauer et al. (2022), which was used during the preceding Sea Spray Chemistry and Particle Evolution (SeaSCAPE) campaign. Waves are generated with a paddle driven by a TEFC (Totally Enclosed, Fan-Cooled) electric motor, up to a maximum height of 0.9 m. During CHAOS, a repeating wave sequence was used which had 2 breaking waves for every 5 wave crests. SOARS features an enclosed air recirculation system with a split duct design above the wave channel where the wind turbines (fans) are located. There are additional (makeup) fans generating positive pressure to reduce mixing of ambient gas and aerosol into SOARS. Airflow through the makeup fans passes through HEPA and activated charcoal filters prior to entering the air ducts upstream of the main fans. A submerged polycarbonate ramp, or "beach", at the end of the channel dissipates residual wave energy and reduces reflected interference within the breaking wave channel. At low wind speeds, HEPA filters and other user-selectable filters (i.e. activated charcoal) can be included in-line with the air stream in the recirculation vents to remove particles and organic compounds. A "tent" constructed of plastic sheeting was built around the paddle during CHAOS to minimize particle or VOC contamination of the SOARS headspace through paddle motion. The tent was positively pressurized with fans forcing air through MERV 8 and potassium permanganate filters.

Water to fill the SOARS channel is sourced from the Pacific Ocean at the nearby Scripps Pier. Seawater is pumped up at the end of the pier from 1-3 m above the sea floor, roughly filtered with an aluminum screen to remove large detritus, and then passes through a rotating drum filter with a variable mesh filter (18-120 μm) to remove phytoplankton (Jio, 2022). Filtered seawater then travels the length of the pier in a gravity flume. Unlike SeaSCAPE, the water volume required to fill the SOARS channel necessitated using the same plumbing and holding tanks as the nearby Birch Aquarium and other SIO labs instead





of pumping water directly out of the gravity flume and transporting by truck to the channel. At the pier entrance, seawater is passed through several additional coarse filters, fed into a large settling tank, and then filtered through high capacity sand filters prior to being pumped into several large holding tanks (Jio, 2022). Finally, the filtered seawater is pumped or gravity-fed

to labs and other facilities. The SOARS channel is filled using either gravity or adjustable-speed water pumps (typically ∼90 gal min$^{-1}$) and optionally passed through additional filters and/or UV-sterilized. During CHAOS, seawater was not filtered or UV-sterilized, and the channel was gravity-filled from the seawater holding tanks. Four separate fills of the SOARS channel were conducted during CHAOS: July 6-18, July 19-21, August 1-12, and August 14-26, 2022. Only data from the two fills in August 2022 are presented in this study due to instrument availability and technical difficulties with the new paddle assembly.

Water temperature can be controlled between -1.6 and 30 °C, and air temperature between -20 and 30 °C. Neither were held constant during CHAOS, and were instead allowed to vary according to the ambient temperature. The channel contains built-in sensors at several locations for measuring air and water temperature, atmospheric $CO_2$ concentration, and water salinity and turbidity. Since the entire SOARS channel is indoors, there are 2 optional lighting mechanisms. Six solar tubes centered on the middle 1/3 of the channel can provide up to ∼6 % of ambient light, which penetrates the full depth of the channel. 40,000 W

of PAR LEDs can provide supplemental lighting during night or stormy conditions.

The SOARS paddle can be programmed to generate wave packets of variable wavelength and amplitude. During CHAOS, two wave packets were superimposed to form 5 wave crests, of which 2 break; this pattern was repeated for the duration of each sampling period. Prior to each sampling period, the headspace air was filtered at a low wind speed to remove particles. Then the wind turbines were set to generate the desired wind speed, and the paddle started to create waves. Occasionally, the

wind turbines were run without the paddle, which can generate SSA at wind speeds higher than ∼ 17 m s$^{-1}$. The wind turbine RPM set points were calibrated using an air velocity meter (TSI Inc. model 9545-A) installed inside the channel with no waves generated. Wind speeds were measured at 0.6 m above the water surface and extrapolated to a value at 10 m ($U_{10}$) following Hsu et al. (1994). Whitecap coverage was calculated from still images collected at high resolution for every measured wind speed. For each wave packet amplitude, there is a single wind turbine set point which generates a whitecap fraction representative of

open-ocean conditions, based on the relationship described in Monahan and Muircheartaigh (1980). During CHAOS, the wave packet amplitude scale was fixed at 1.3, which yields an open-ocean equivalent whitecap coverage at a wind turbine set point of 1550 rpm, corresponding to an extrapolated $U_{10}$ of 18.5 m s$^{-1}$. Measurements collected during CHAOS were made at wind turbine speeds of 850, 1200, 1400, 1500, 1600, and 1800 rpm, which correspond to $U_{10}$ of 9.57, 13.84, 16.28, 17.50, 18.72, and 21.16 m s$^{-1}$, respectively. Measurements made at 1600 rpm (18.72 m s$^{-1}$) are considered to represent open-ocean breaking

wave conditions, through comparison with Monahan and Muircheartaigh (1980). For all other wind speeds measured during CHAOS, the fixed wave amplitude meant the whitecap coverage is not comparable to equilibrium open-ocean conditions and only the relative influence of wind speed alone can be assessed.

## 2.2 Ice Nucleating Particle Measurements

Ice nucleating particle measurements were conducted at all wind speeds. A Colorado State University (CSU) Continuous Flow

Diffusion Chamber (CFDC; Section 2.2.1) was used to capture online measurements at high temporal resolution (∼15 min),



and aerosol filter samples were collected and subsequently analyzed with the CSU Ice Spectrometer (IS; Section 2.2.2) to provide INP temperature spectra down to -30 °C. Chemical pre-treatments of aerosol filter suspensions allowed INPs produced in SOARS to be classified by broad composition (Section 2.2.3), and Atomic Force Microscopy was used to assess INP morphology and phase state (Section 2.2.4). Water samples were collected daily from SOARS, and seawater ice nucleating
entity (INE) temperature spectra were also measured using the IS as a complement to the aerosol results (Section 2.2.2). CFDC measurements (Section 2.2.1) presented here exclude the first 15 minutes of each sampling period to allow particle concentrations to reach an approximate steady-state. IS filters (Section 2.2.2) were started ∼15 min into each sampling period for the same reason.

### 2.2.1   Continuous Flow Diffusion Chamber

Real-time measurements of INP concentration were collected using a CSU Continuous Flow Diffusion Chamber (CFDC), a vertically oriented, ice-thermal diffusion chamber (Rogers, 1988; Rogers et al., 2001; DeMott et al., 2015). The HIAPER (CFDC-1H) version of the CFDC used during CHAOS has been previously described in detail and will only be briefly discussed here (e.g. McCluskey et al., 2018a; Moore, 2020; DeMott et al., 2023; Moore et al., 2024). Prior to entering the top of the CFDC chamber, the sample aerosol stream drawn from SOARS was dried to below the frost point with diffusion driers, then passed
through two sequential single-jet impactors (50 % aerodynamic diameter cut size $D_{50}$=2.4 μm) to remove large aerosols. Within the chamber, particles are first exposed to near steady-state humidity and temperature conditions conducive to the activation of cloud droplets and ice crystals, followed by a water-subsaturated region to evaporate haze and cloud droplets back to aerosol sizes. Ice crystals are then detected optically at the base of the chamber using an optical particle counter (OPC) and distinguished by size from aerosols and any remaining cloud droplets (Barry et al., 2021b). The upper region of the chamber
was held under water supersaturated conditions (typically 104 % to 108 %) for this campaign to emphasize the immersion freezing mode of ice nucleation and give comparable results to offline techniques (DeMott et al., 2016, 2017, 2018; Barry et al., 2021b).

The aerosol lamina temperature was held at -25 °C or -30 °C during CHAOS to maximize the instrumental signal-to-noise ratio and accommodate limited sampling durations at each wind speed. Paired measurements of the sample air stream (10 min)
and HEPA-filtered air (5 min) were used to quantify instrument noise (DeMott et al., 2017). All measurements presented here have been corrected for CFDC background using adjacent filtered-air periods, as in Moore (2020) and Barry et al. (2021b). This correction is achieved using a Poisson model incorporating the detection rates of INPs during ambient and filtered-air measurements. Confidence intervals on INP concentrations and statistical differences between sample and filtered-air periods are assessed at the same time as the background correction, and follow Krishnamoorthy and Lee (2012). All concentrations are
converted to standard conditions to allow for direct comparisons between measurements at varying temperatures (STP; 0 °C and 100 kPa).

Nucleated ice crystals were collected for offline analysis following the OPC at the base of the CFDC chamber and analyzed using Atomic Force Microscopy to ascertain differences in INP morphology and phase state with wind speed (Section 2.2.4).



Ice crystals were collected onto substrates using a single-jet impactor with a 50 % cut-size of 4 μm aerodynamic diameter
(McCluskey et al., 2014; Barry et al., 2021b).

### 2.2.2   Ice Spectrometer Measurements

Aerosols produced in SOARS were collected onto pre-cleaned 0.2 μm pore size, 47 mm diameter track-etched polycarbonate
membrane filters (Whatman Nuclepore) in pre-sterilized aluminum inline filter housings (Pall), using the protocols described
in Barry et al. (2021a). Sample flow rates were held at ∼5 std lpm (slpm) and the sample stream passed through a silica gel
diffusion drier prior to particle collection to prevent saturation/wetting of the filters. Filter collection volumes ranged from 182
to 855 std L, with the higher volumes representing longer sampling durations at lower wind speeds to increase particle mass.
Blank filters were collected regularly by installing filters in housings and connecting to the same tubing used for SSA sampling,
without airflow. Seawater was sampled from either the rear end of the SOARS channel (beach) or underneath the aerosol
sampling manifold, approximately halfway up the water column, using a peristaltic pump and silicone tubing to minimize cell
rupture for biological measurements. Filters and seawater were either analyzed immediately or stored frozen (-20 °C) prior to
analysis.

   Offline measurements of INP and INE immersion freezing temperature spectra were made using the CSU Ice Spectrometer
(IS), which has been comprehensively described in its present form elsewhere (Hiranuma et al., 2015; DeMott et al., 2018;
Hill et al., 2023). Aerosol filters were re-suspended in 8 mL of 0.1 μm filtered DI water, then 50 μL aliquots of either seawater
or aerosol suspensions were dispensed into sterile 96-well PCR trays (Optimum Ultra, Life Science Products). Dilutions of
each sample were used to extend the measurement temperature range; these were made in 0.1 μm filtered DI water for aerosol
filter suspensions and 0.1 μm filtered artificial seawater (NeoMarine, Brightwell Aquatics) for seawater samples. The trays
were then placed into temperature-controlled aluminum blocks inside the IS, and cooled at ∼0.33 °C min$^{-1}$. Freezing events
were detected optically from CCD camera images collected at 1 Hz. A 0.1 μm filtered DI water or artificial seawater negative
control was included with each IS measurement and used to correct sample results for INPs present in the water used for
resuspension and dilution. INP concentrations in the aerosol suspensions or seawater were calculated following Vali (1971),
then converted to concentrations in SOARS headspace air for aerosol filters (reported at STP; 0 °C and 100 kPa). Confidence
intervals were derived following Agresti and Coull (1998), and the LOD determined as in Moore et al. (2024). The average
background number of INPs from the collected blank filters (4) were used to adjust filter sample concentrations; measurements
are not reported if blank-corrected values fell below zero (Moore et al., 2024). Temperature spectra of seawater samples have
been adjusted by +2 °C to account for freezing point depression due to salinity.

### 2.2.3   Chemical Composition of INPs in the Ice Spectrometer

Inferences about INP composition are possible from pre-treatments of aerosol filter suspensions or seawater prior to analysis
with the IS. Heat treatments are used to assess the contribution of biological INPs to a total sample population (Hill et al., 2016;
Suski et al., 2018), as INPs produced by fungi and bacteria are often proteinaceous (Pummer et al., 2015) and denatured by
heating. Aliquots of either re-suspended particles from aerosol filters or seawater were immersed in boiling water for 20 min




before being cooled to room temperature and then analyzed with the IS as normal (Section 2.2.2). The difference between the pre- and post-heat treated sample represents the biological INP contribution. The proportion of refractory, typically mineral, INPs are identified through oxidation experiments that remove organic material (Suski et al., 2018; McCluskey et al., 2018c).

Sample aliquots are digested for 20 min with 10 % hydrogen peroxide while immersed in boiling water, with two UVB fluorescent bulbs (Exo Terra) illuminating the samples to generate hydroxyl radicals. After cooling, catalase (MP Biomedicals, PN 100429) is added to remove any excess hydrogen peroxide and prevent significant freezing point depression (Suski et al., 2018). The INP temperature spectrum remaining after oxidation is inferred to be the mineral (or other inorganic) component, and the difference between pre-and post-oxidation spectra corresponds to organic INPs.

### 230    2.2.4    Single Particle Atomic Force Microscopy of INPs

INPs collected in the CFDC were deposited onto hydrophobically coated (Rain-X) silicon substrates (Ted Pella, Inc.) and stored in clean Petri dishes inside a laminar flow hood (NuAire, Inc., NU-425-400) at ambient temperature (20-25 °C) and pressure prior to analysis (Lee et al., 2020; Kaluarachchi et al., 2022a, b). Samples collected at four wind speeds (9.57, 16.28, 18.72, and 21.16 m s$^{-1}$) were analyzed to assess the distribution of physicochemical properties under varied wind stress. A molecular force

probe 3D AFM (Asylum Research, Santa Barbara, CA) was used to image individual INPs at ambient temperature (20-25°C) and pressure, as described in prior studies (Ray et al., 2019; Lee et al., 2020). A custom humidity cell was used to control RH between 20 % and 60 %. Prior to AFM measurements at a particular RH, samples were allowed to equilibrate for at least 10 minutes to ensure thermodynamic equilibrium with the surrounding water vapor (Lee et al., 2017, 2020; Madawala et al., 2021). Silicon nitride AFM tips (MikroMasch, model CSC37, tip radius of curvature ∼10 nm, nominal spring constant 0.5-0.9

N m$^{-1}$) were used for AFM imaging and force spectroscopy measurements (Lee et al., 2017; Madawala et al., 2021). AFM AC (intermittent contact) imaging mode was used to collect 3D height images of individual INPs to determine their morphology, and to quantify their volume-equivalent diameter, as described previously (Ray et al., 2019; Kaluarachchi et al., 2022b). For morphological analysis, approximately 50 individual particles were studied for each sample, with volume-equivalent diameters ranging from 0.05 – 1.0 μm. Particles were classified into six main types: rounded, core-shell, prism-like, rod, aggregate and

irregular. Example images of particles at 20 % RH in each category are shown in Fig. A2.

Organic particle phase state was identified for samples at 20 % and 60 % RH, as in previous studies (Lee et al., 2017, 2020). These RH values were selected as benchmarks based on previous phase state studies on sucrose that showed solid-to-semisolid and semisolid-to-liquid phase transitions at ∼20 % and 60 % RH, respectively (Lee et al., 2017; Ray et al., 2019; Madawala et al., 2021). Briefly, AFM force spectroscopy (i.e., force plots), was performed on individual core-shell particles at a particular

RH by probing within the shell region of each particle. At least five force plots were collected for each individual particle at both 20 % and 60 % RH, with a maximum force of 20 nN and scan rate of 1 Hz. The viscoelastic response distance (VRD) and relative indentation depth (RID), or ratio of the indentation depth to the particle height, were then quantified, which can be related to the viscosity of the material (Lee et al., 2020; Kaluarachchi et al., 2022a). A previously reported framework based on VRD and RID measurements was then utilized to identify the phase state of each particle at 20 % and 60 % RH (Lee et al.,



2017). A total of 5, 19, 12, and 13 individual core-shell particles were studied for the 9.57, 16.28, 18.72, and 21.16 m s$^{-1}$ wind speed conditions, respectively (Table A1).

Since the total number of individual particles that can be realistically studied with AFM is somewhat limited, a probability distribution analysis to assess the statistical significance of the AFM results was employed (Cappa et al., 2021, 2022; Kaluarachchi et al., 2022b). Briefly, the probability distribution curves associated with the likelihood of sampling one of the

six particle morphology types, or one of the three phase states, were generated using a Markov chain Monte Carlo method for a "true" population of 10,000 particles. The resulting distributions were fit with Gaussians to provide standard deviation estimates for both morphology and phase state measurements.

## 2.3 Aerosol Size Distribution Measurements

Several dedicated instruments were used to measure aerosol size distributions during CHAOS, using different aerosol inlet

configurations. All aerosol streams were dried with silica gel diffusion driers prior to measurement to below the efflorescence relative humidity of sea salt, ∼45-48 % (Tang et al., 1997). The first set of measurements used in this study consist of a TSI Scanning Mobility Particle Sizer (TSI, SMPS 3936) for aerosols in the range 14-750 nm and a TSI Aerodynamic Particle Sizer (TSI, APS 3321) for particles between 0.5-20 μm. The SMPS and APS sampled from a 3/8 inch diameter stainless steel inlet that entered the side of the SOARS channel and then turned 90° to face into the air flow. It was located approximately 0.6 m

above the water surface and angled roughly 45° below horizontal, towards the water's surface. The INP filter measurements were made with a similar inlet located 2-3 m further down the channel, but oriented parallel to the water's surface. A second set of aerosol measurements were collected with a Scanning Electrical Mobility Analyzer (BMI, SEMS model 2002) between 10 and 1340 nm (mobility diameter) and another Aerodynamic Particle Sizer (TSI, APS model 3321), both of which sampled behind a 2.5 μm cyclone. The SEMS and APS data were merged at 650 nm after converting the APS from aerodynamic to

mobility diameter assuming a particle density of 2.0 g cm$^{-1}$. The SEMS and APS sampled from a shared aerosol manifold with the CFDC, which had a vertically oriented 1/2 inch diameter stainless steel inlet that entered from the top of the SOARS channel and sampled ∼0.6 m above the water surface. Theoretical particle transmission efficiency calculations were performed for both sets of inlets (INP filter/SMPS + APS and CFDC/SEMS + APS) and are shown in Fig. A3 as a function of the SOARS fan speed. These calculations were performed in aerodynamic diameter with a particle density $\rho$=1 and later corrected for

expected particle density, water uptake, and shape factor following Tang et al. (1997) and Zieger et al. (2017). Significant vibrations and vertical movement of the INP filter and SMPS + APS sampling inlets were observed at higher wind speeds, with unknown effects on particle line losses that are not accounted for in these theoretical calculations.

These particle measurements were primarily used to normalize the INP concentrations, as described in Section 3.1. Particle surface area and volume distributions were calculated for each number distribution assuming particle sphericity, as were num-

ber concentrations of particles larger than 500 nm dry diameter (n500). Due to the differences in expected aerosol transmission (Fig. A3) between the horizontally and vertically oriented inlets, the SMPS + APS data was used to normalize the INP filter results. The SEMS + APS observations were intended to correct the CFDC data, however, SEMS data was only available for the second half of August. So instead, data from the OPC at the base of the CFDC chamber, which is limited to particles larger





than ~300 nm, was used to provide aerosol concentrations to normalize the CFDC INP measurements. Correction factors for
total particle number, n500, surface area, and volume concentrations were derived for the CFDC OPC based on the available
SEMS data (Fig. A4) and applied to the OPC observations shown here.

CFDC operation requires the incoming air stream to be dried to below the frost point at the given measurement temperature
(typically -25 °C or -30 °C), so aerosols enter the CFDC at dry sizes. However, particles will deliquesce, and some will activate
into cloud droplets under the water supersaturated conditions present in the top section of the CFDC chamber. Any particles
not activated into ice crystals will evaporate in the water-subsaturated region at the bottom of the chamber (Section 2.2.1),
which is held at ice saturation. Following Murphy and Koop (2005), the saturation vapor pressures with respect to ice and
water were calculated during each period based on the measurement temperature, as well as the resulting RH. Dry particle
sizes were estimated assuming spherical, sea salt particles with a hygroscopic growth factor (HGF) of 1.7 for the 70-75 % RH
range calculated (Zieger et al., 2017). The CFDC OPC was calibrated against polystyrene latex spheres (PSLs) and glass beads
of known sizes and refractive indices, and size distributions calculated assuming a refractive index of n=1.5 for sea salt (Tang
et al., 1997).

## 3   Results and Discussion

### 3.1   SSA and INP production at Varying Wind Speeds

Measurements of both SSA and INPs were made at six $U_{10}$ wind speed equivalents (9.57, 13.84, 16.28, 17.50, 18.72, and 21.16
m s$^{-1}$) during CHAOS. Normalized histograms of integrated particle number, number >500 nm diameter (n500), surface area,
and volume concentrations are shown in Fig. 1 for all measured wind speeds from the corrected CFDC OPC measurements, and
example size distributions from the SEMS + APS and SMPS + APS in Fig. A5 (Sec. 2.3). As expected from numerous previous
measurements (e.g. Lewis and Schwartz, 2004; O'Dowd and de Leeuw, 2007; de Leeuw et al., 2011), particle concentrations
generally increased with wind speed in the SOARS channel. Large variability in each aerosol metric was observed at all
wind speeds (Fig. A5), with a clear increase in aerosol concentration between the 16.28 and 17.50 m s$^{-1}$ wind speeds. This
variability occurred for measurements collected both days or weeks apart and on the same day, if wind speeds were repeated,
and the source is unknown (see Fig. A5). A previous study of wind profiles in a wind-wave tank (Vollestad and Jensen, 2021)
found that while the horizontal wind speed displayed the expected, approximately logarithmic profile, secondary flows due to
the confined channel were found to impact the observed vertical velocity structure. Modification of the near-surface wind and
turbulence due to the presence of waves has been observed in wind-wave tanks (Zavadsky and Shemer, 2012; Villefer et al.,
2021) and in models (Chen et al., 2019), and varies with the fetch (Lamont-Smith and Waseda, 2008), as well as the presence
of swell in addition to wind-waves (Villefer et al., 2021). Variation in secondary flow structure is a possible explanation for
some of the variability seen in particle concentrations at the same nominal wind speed during CHAOS.

The maximum observed values for particle number, surface area, and volume were much larger during CHAOS than South-
ern Ocean values (Moore et al., 2022), by factors of ~50, ~45, and ~7, respectively. At least some of these differences are
likely a result of the differences in time scale and fetch, with open ocean measurements closer to steady state, and integrated





over a larger area with potentially more variability. Additionally, the SOARS channel is a closed system where horizontal and vertical SSA fluxes are suppressed, allowing particle concentrations to build until losses are equal to emissions. Size distribution measurements (Fig. A5) suggest the size distribution shape and mode size is similar across wind speeds in SOARS, but

with larger variability in number concentration at higher wind speeds, particularly in the accumulation mode. At wind speeds below 18.5 m s$^{-1}$, the fixed 1.3 amplitude-scaled waves generated by the SOARS paddle led to higher whitecap coverages than would be anticipated in the open ocean for equilibrium conditions, and for the highest, 21.16 m s$^{-1}$, whitecap coverage was lower than open ocean values (Monahan and Muircheartaigh, 1980). This likely led to an overestimation of particle production at low wind speeds and underestimation at the highest. Additional tests are currently underway to study particle production

when the wave amplitude is varied along with the wind speed to match open ocean whitecap fractions, which may reduce some of the large observed variability in particle production at some wind speeds during CHAOS.

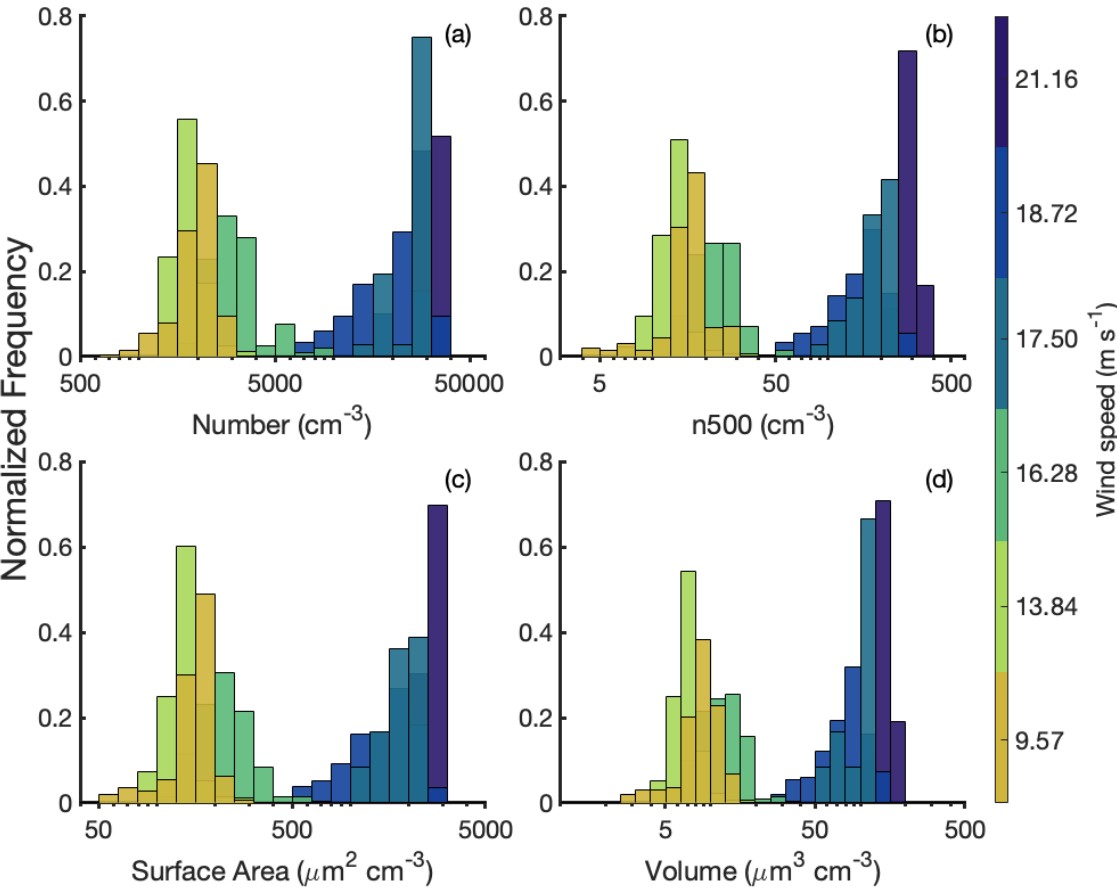

**Figure 1.** Normalized frequency distributions of particle (a) number, (b) number >500 nm diameter (n500), (c) surface area, and (d) volume concentrations at each measured wind speed during CHAOS.



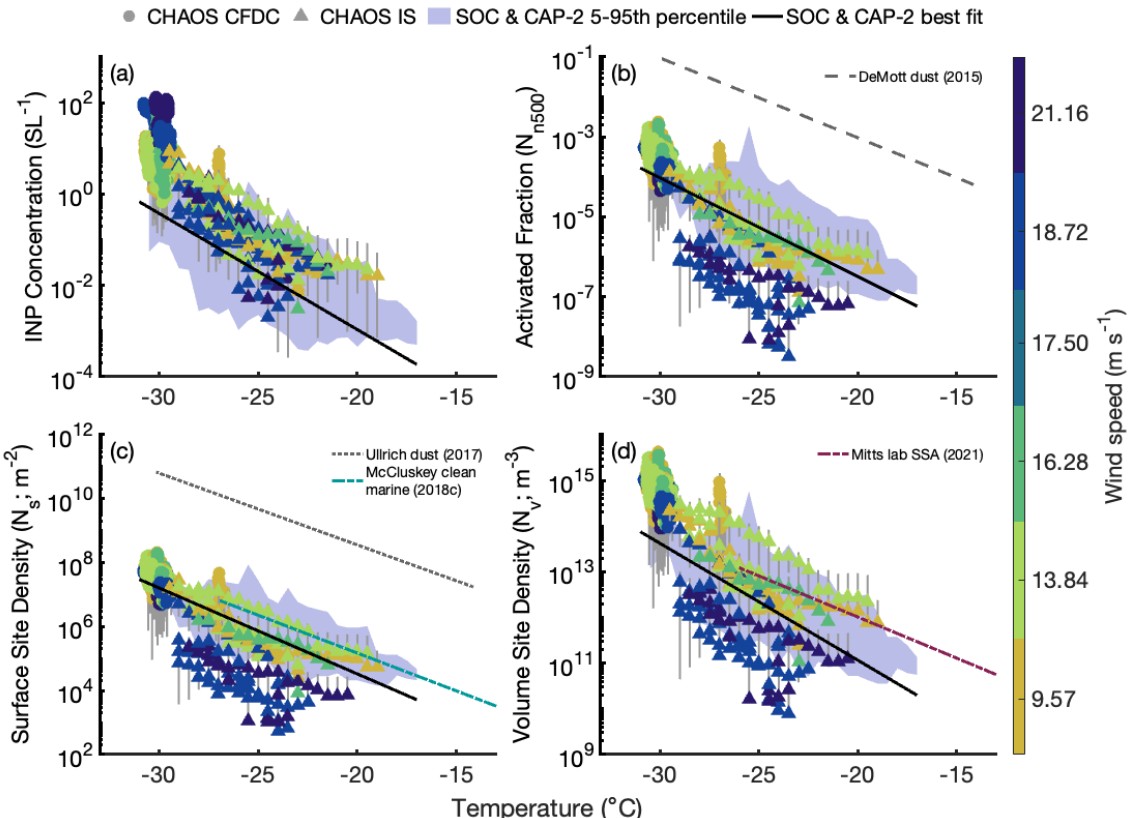

**Figure 2.** INP (a) number concentration, (b) normalized by n500 ($N_{n500}$), (c) normalized by aerosol surface area ($N_s$) and (d) normalized by aerosol volume ($N_v$) temperature spectra during CHAOS. CFDC measurements are indicated by circles and IS filter observations by triangles; both are colored by the wind speed during each measurement period. The purple shading in each panel indicates the 5th-95th percentile of values observed in the marine boundary layer during SOCRATES and CAPRICORN-2 (Moore et al., 2024), and solid black lines are the best-fit lines for each variable from these campaigns. In (b), the grey dashed line shows the DeMott et al. (2015) parameterization for dust based on n500, using the median n500 value measured during 18.72 m s$^{-1}$ wind speed periods. In (c), the grey dotted line indicates the Ullrich et al. (2017) parameterization for dust $N_s$, and the blue dot-dash line shows the $N_s$ parameterization from McCluskey et al. (2018c) for North Atlantic clean marine air. The dashed magenta line in (d) indicates the Mitts et al. (2021) lab-based parameterization for marine $N_v$.

A summary of the INP results from CHAOS, along with relevant model parameterizations, are displayed in Fig. 2, which shows INP measurements from the CFDC (Section 2.2.1) and IS filters (Section 2.2.2) as a function of temperature. Similar observations made in the Southern Ocean marine boundary layer during the Southern Ocean Cloud Radiation Aerosol Transport Experimental Study (SOCRATES, hereafter SOC) aircraft campaign and the second Clouds, Aerosols, Precipitation, Radiation and atmospherIc Composition Over the southeRN ocean (CAPRICORN-2, hereafter CAP-2) ship campaign are shown in each panel in the light purple shading (Moore et al., 2024). Figure 2a shows measured INP concentrations, while the other panels show different normalizations commonly used in models (Fig. 2b-c) or suggested for marine INPs (Fig. 2d). Figure 2b displays



INP concentrations normalized by n500 ($N_{n500}$), which has been used previously for dust (DeMott et al., 2015) and biological

INPs (Tobo et al., 2013) due to observed relationships with supermicron aerosol. Figure 2c is normalized by aerosol surface area, which has been widely used for multiple INP types, including marine INPs (Niemand et al., 2012; Ullrich et al., 2017; McCluskey et al., 2018c). Normalization by aerosol volume (Fig. 2d) was suggested by Mitts et al. (2021) for marine INPs on the basis of laboratory experiments, but measurements from the Southern Ocean (Moore et al., 2024) did not support a similar relationship for ambient data, and nor do these measurements from CHAOS.

INP concentrations and normalized values vary in their consistency with CAP-2 and SOC measurements (Fig. 2), which themselves agreed well with previous observations from the Southern Ocean and mid-latitude North Atlantic (McCluskey et al., 2018a; Schmale et al., 2019; Tatzelt et al., 2022; Moore et al., 2024). In general, CHAOS measurements at wind speeds <17 m s$^{-1}$ agree with those from SOC and CAP-2 and those at higher wind speeds do not, although there are some differences between CFDC and IS observations that will be discussed more below. INP concentrations during CHAOS were on the high

end (above the 50th percentile) of Southern Ocean values, and CFDC ($\leq$-27 °C) measurements at wind speeds above 17 m s$^{-1}$ are above the 95th percentile of CAP-2 and SOC values by about an order of magnitude. As anticipated, the DeMott et al. (2015) n500-based parameterization (Fig. 2b) and Ullrich et al. (2017) $N_s$ parameterization (Fig. 2c) for dust INPs overestimate CHAOS values by several orders of magnitude. $N_{n500}$ and $N_s$ measured by the CFDC during CHAOS overlap with Southern Ocean observations, though are biased high (Fig. 2b-c). IS $N_{n500}$ and $N_s$ values for wind speeds <17 m s$^{-1}$ are within the

5th-95th percentile of CAP-2 and SOC values, while those at high wind speeds are almost entirely below the 5th percentile. Interestingly, the agreement for $N_v$ is better overall, although CFDC measurements are all above the 50th percentile and extend above the CAP-2 and SOC 95th percentile, while IS measurements at higher wind speeds fall below the 5th percentile (Fig. 2d). Similarly to the CAP-2 results (Moore et al., 2024), the Mitts et al. (2021) $N_v$ parameterization has a lower slope than the CHAOS dataset, and is near the upper bound of measured values at all temperatures. Variable agreement among $N_{n500}$, $N_s$, and

$N_v$ suggests a different shape to the particle size distribution in the SOARS channel than the Southern Ocean MBL, since all of the aerosol concentrations are enhanced in SOARS relative to ambient measurements (Fig. 1), but only $N_v$ has a similar range as ambient observations. This is supported by example size distributions from CHAOS, which show enhancements in aerosol concentrations between 0.1 - 1 μm relative to CAP-2 distributions, with the discrepancy increasing with wind speed (Fig. A5).

        CFDC INP concentrations (circles in Fig. 2, $\leq$-27 °C) generally increase with wind speed, while variability is reduced

following normalization by aerosol concentrations, as expected if INPs are emitted proportionally to SSA. The reduction in spread after normalization is shown even more clearly in the time series of CFDC data presented in Fig. A6. However, it is also clear from Fig. A6 that on some days, INP concentrations were the same up to a wind speed threshold of ~17 m s$^{-1}$ (8/8/22, 8/17-8/19/22). Other days did not sample enough wind speeds to assess this variation. This agrees with what was observed for SSA concentrations in Fig. 1, which showed a distinct increase in aerosol concentrations between the 16.28 and 17.50 m s$^{-1}$

wind speeds.

        On the other hand, INP concentrations measured from the aerosol filters (triangles in Fig. 2, $\geq$-28 °C) did not have a clear relationship with wind speed. This difference may be due to the different averaging times of the CFDC (~5 minutes) versus the IS filters (2-3 hr), differences in inlet orientations or locations (Sec. 2.3), or differences in the aerosol sampled.



The CFDC sampled ∼2 m upstream of the filters, with a vertically oriented inlet, whereas the IS filters used a horizontal inlet
facing into the wind. Despite the anticipated enhancements in particle transmission ∼1 μm for the IS filter inlet at higher wind
speeds and otherwise similar efficiencies to the CFDC inlet (Fig. A3), the consistently higher concentrations measured by the
CFDC at the same wind speed suggest particle losses in the IS filter inlet may not be accurately captured by these theoretical
calculations. Future studies should make both online and offline measurements on the same or more similar inlets to reduce
these uncertainties. The IS and CFDC are also largely measuring INPs at different temperatures, with the CFDC primarily
targeting INPs active ∼ -30 °C and the IS sensitive to INPs at warmer temperatures. Temperature-dependent differences in
INP composition may also be driving the observed discrepancy between the IS and CFDC results, especially if emission of
different types has contrasting dependencies on wind speed.

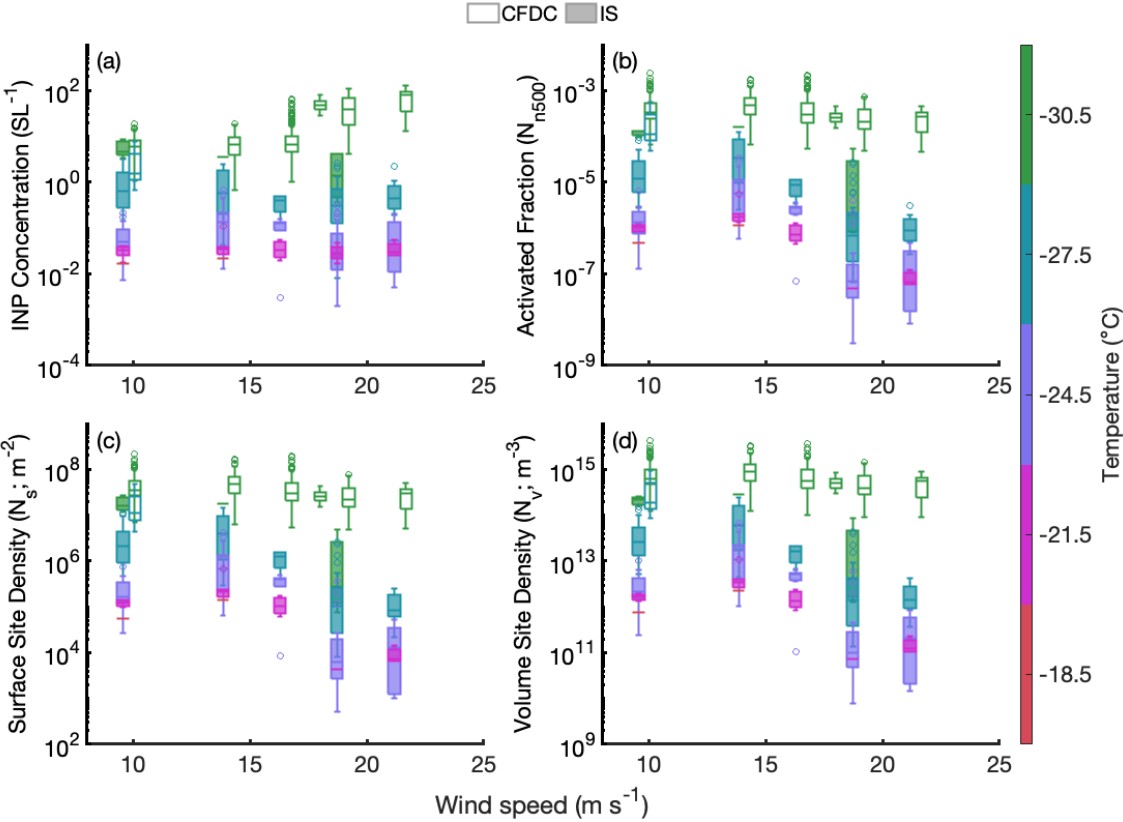

**Figure 3.** Box plots of observed (a) INP concentration, (b) $N_{n500}$, (c) $N_s$ and (d) $N_v$ as a function of wind speed during CHAOS. Observations
are separated into 3 °C temperature bins (indicated by color), with CFDC measurements shown as open boxes and IS filter data as shaded
boxes. CFDC data are offset to the right by +0.5 m s$^{-1}$ for clarity.

Normalized INP concentrations for both instruments generally decreased with increasing wind speed, especially above
∼17 m s$^{-1}$, although decreases were more modest for the CFDC than the IS. This can be seen more clearly in Fig. 3, which
displays the same data as Fig. 2 as a function of wind speed in 3 °C temperature bins, with CFDC and IS filter ranges





indicated by box plots. Also clear in Fig. 3 is the large inter-sample variability observed during CHAOS for measurements collected at similar wind speeds and temperatures. INP concentrations in the Southern Ocean MBL were found to increase with wind speed, and to retain the same wind speed dependence after normalization by aerosol number, surface area, and volume (Moore et al., 2024). Even if only considering the CFDC observations, normalized INP concentrations have a small

but negative relationship with wind speed during CHAOS. One possible explanation is that loss mechanisms such as dry and wet deposition have lower rates in SOARS, where aerosol was sampled from 0.6 m above the water surface, compared to the ambient marine boundary layer, where measurements were collected from 18.4 m above sea level on the ship and ∼150 m on the aircraft during CAP-2. This would alter the particle size distributions in SOARS, especially at larger sizes where loss rates are higher. As discussed earlier, higher concentrations were seen in the accumulation mode during CHAOS than CAP-2

(Fig. A5). Unfortunately, losses at larger sizes are hard to assess with the available size distribution measurements since the SEMS + APS sampled behind a 2.5 μm cyclone and the SMPS + APS had an inlet similar to the IS (Sec. 2.3) and thus likely also experienced additional losses not accounted for in the theoretical calculations. Overall, the results from CHAOS may be more representative of interfacial fluxes rather than marine boundary layer or cloud-base values. As previously discussed in relation to measured particle concentrations, the fixed 1.3 amplitude scaling for wave height used during CHAOS may also be

obscuring the true INP-wind speed relationships, which requires further measurements with co-varying wave amplitude and wind speed to resolve. Seawater INE concentrations were relatively stable throughout CHAOS (Fig. A7) and agree well with previous measurements from the Scripps Pier, as well as the North Indian Ocean (Beall et al., 2022) and mid-Atlantic (Gong et al., 2020), and are higher than observations from the Southern Ocean (McCluskey et al., 2018a) or Barents Sea (Hartmann et al., 2021) by 1-2 orders of magnitude. The INE stability across multiple fills of the SOARS channel and over time with the

same water indicates the observed INP-wind speed relationships were driven by wind-wave interactions rather than biological activity in this experiment.

### 3.2 INP Composition and Phase State Changes under Increasing Wind Speeds

The fractional composition of INPs (Section 2.2.3) as a function of wind speed is shown in Fig. 4 for three temperature ranges: -19 to -23 °C, -23 to -26 °C, and -26 to -29 °C. Composition data is only reported when the treated and un-treated sample were different at the 95 % confidence level, and the fraction of data not meeting this criteria are shown in Fig. A8 as a function

of temperature. The generally low fractions of heat treated spectra that significantly differed from the base spectra (green dots in Fig. 4c,f,i) indicate the collected INPs were largely unaffected by heat treatments, although consistently high biological fractions (∼1) were observed at temperatures >-23 °C and wind speeds below 15 m s$^{-1}$ (Fig. 4a-c). Low wind speeds (<∼13 m s$^{-1}$) may favor enrichment of biological INPs in the sea surface microlayer (Wilson et al., 2015; Engel et al., 2017; Irish

et al., 2017; Ickes et al., 2020; Hill et al., 2023), which is consistent with this result. On several days, especially at the end of the second water fill (8/16/22-8/19/22), heat treatments led to increased INP concentrations over the untreated filters at temperatures of -23 to -26 °C, which are shown as biological fractions >1 in Fig. 4d-e, especially at the highest wind speeds. This observation is uncommon but was observed by McCluskey et al. (2018b) during a laboratory-simulated phytoplankton bloom grown in a Marine Aerosol Reference Tank (MART; Stokes et al., 2013). It was suggested to be a result of lysis of



microbial cells upon heating, releasing IN-active material, or the dissolution and redistribution of organic material between particles, leading to a net increase in the number of particles with IN-active organic material. This contrasts with the consistent decrease after heating also presented in McCluskey et al. (2018b) for a phytoplankton bloom grown in the SIO glass channel during the IMPACTS (Investigation into Marine Particle Chemistry and Transfer Science) campaign (Wang et al., 2015), where larger proportions of biological INPs were inferred to be released in response to increased seawater biological activity.

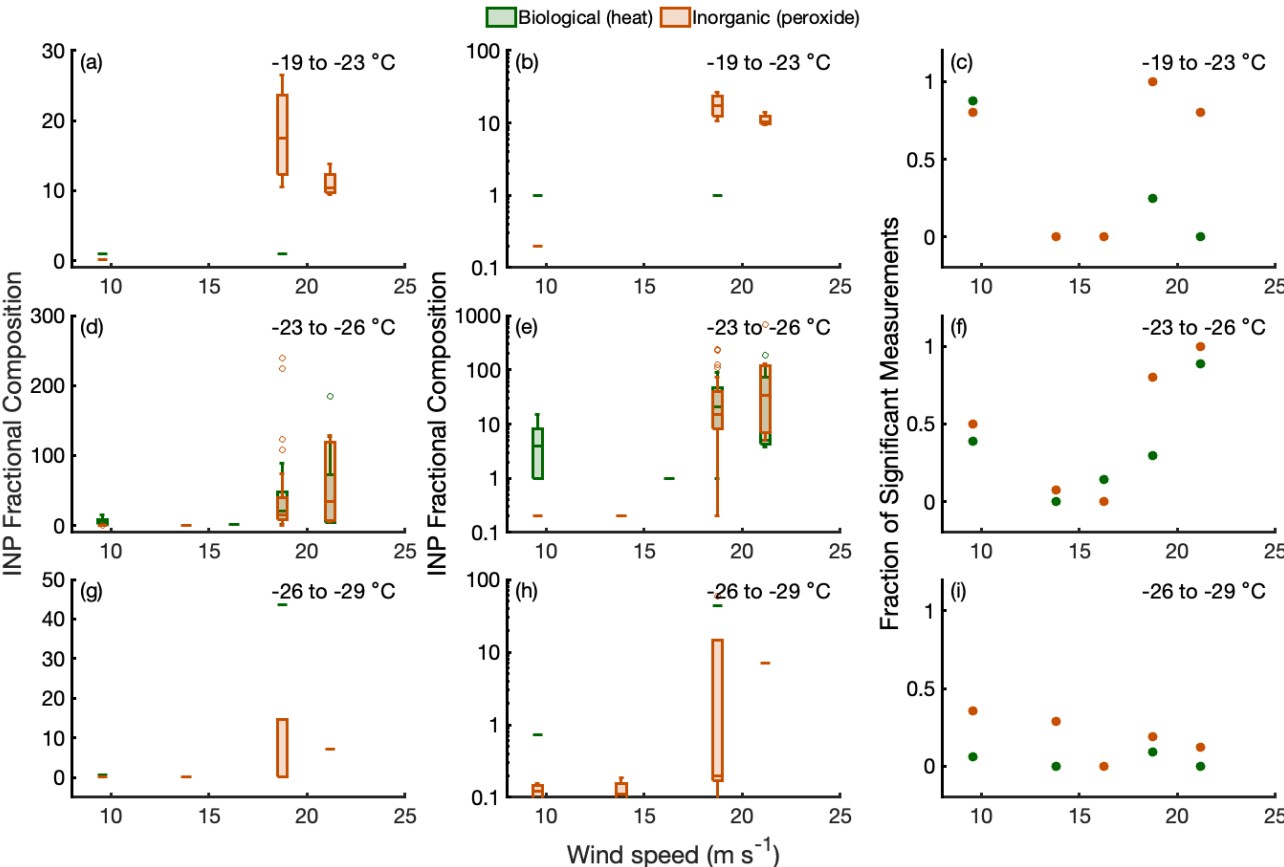

**Figure 4.** Box plots of biological (green) and inorganic (orange) INP fractional composition as a function of wind speed for IS filter measurements at (a) -19 to -23 °C, (d) -23 to -26 °C, and (g) -26 to -29 °C. Panels (b), (e), and (h) are identical to (a), (d), and (g), respectively, except with a log y-axis so smaller values are visible; zero values are plotted at a fixed value of 0.2 on the log axes. Only treatments that differ from the base spectra at the 95% confidence level are included in (a-b), (d-e), and (g-h). Panels (c), (f), and (i) indicate the fraction of measurements meeting this criteria as a function of wind speed and at temperatures of -19 to -23 °C (c), -23 to -26 °C (f) and -26 to -29 °C (i).

At low wind speeds (<15 m s$^{-1}$) and below -23°C, heat-stable organic INPs (low biological and low inorganic fractional composition) were the dominant INP type, corresponding to the DOC-INP type described in McCluskey et al. (2018b). This is in accordance with a number of laboratory (McCluskey et al., 2018b) and field (Rosinski et al., 1987; Wilson et al., 2015;



Ladino et al., 2016; Alpert et al., 2022) measurements, although other studies have inferred the dominance of proteinaceous or heat-labile material (Knopf et al., 2011; Wang et al., 2015; Irish et al., 2017). By contrast, at higher wind speeds, inorganic

or refractory INPs were the dominant type observed at all temperatures. At wind speeds >15 m s$^{-1}$, nearly all peroxide-treated filter samples had higher INP concentrations than the untreated samples (inorganic fractional composition >1 in Fig. 4a-b, d-e, g-h), and many of these corresponded to the heat-treated samples with enhanced INP concentrations described above. All of the samples with enhanced concentrations following peroxide digestion had a characteristic shape to their temperature spectra, an example of which is shown in Fig. A9. In contrast to the typical log-linear marine INP spectra (DeMott et al., 2016), dramatic

increases are seen in peroxide-treated results at warm temperatures, which flatten out ∼-23 °C and meet or approach the untreated spectra around -27 °C. This is reminiscent of INP temperature spectra identified as biological (Hill et al., 2016; Suski et al., 2018), which have large warm temperature INP populations which are reduced to log-linear spectra following heating and/or peroxide digestion, only inverted. An increase in INP concentration after peroxide digestion has not been reported before for marine INPs, but is hypothesized to be the result of enhanced release of large particles at high wind speeds in SOARS,

which may contain multiple INPs. The production of spume droplets through the tearing of wave crests, which produces particles predominantly >10 μm and is increasingly active for wind speeds exceeding ∼9 m s$^{-1}$ (Monahan et al., 1986; Sofiev et al., 2011), is the most likely mechanism consistent with the observed wind speed dependence. The atmospheric lifetime of such particles is very short, which may explain why this has not been observed in ambient measurements or laboratory studies with low wind speeds. Organic material in seawater, including carbohydrates, lipids, and proteins, are well known to self-

assemble into microgels which can range in size from <10 nm (single macromolecule) to μm-sized colloidal gels (Chin et al., 1998; Verdugo, 2012). INPs could be trapped in this gel matrix, emitted as large spume drops, and then released following the breakdown of the organic material during peroxide digestion. If so, the composition of the INPs themselves cannot be inferred from these results, since they could be either inorganic contaminants (dust) which are stable against peroxide digestion, or heat-stable organics which the 20-min digestion used here is not long enough to both release from their gel matrix and break

down.

    Additional information about the composition of INPs produced in SOARS was provided by AFM analysis of submicron ice crystal residuals collected in the CFDC (Section 2.2.4). Six particle categories were identified based on 3D height images of particles collected at 4 wind speeds (9.57, 16.28, 18.72, and 21.16 m s$^{-1}$): rounded, core-shell, prism-like, rod, aggregate and irregular (Fig. A2). These are similar to the categories identified for ice crystal residuals during SeaSCAPE (DeMott et al.,

2023), except rod and irregular particles were not identified during SeaSCAPE. Some of the particles in the rod and irregular classes are morphologically similar to known contaminants from the SOARS channel itself, and these particle classes will not be further discussed here. Prism-like particles did not display a clear relationship with wind speed (Fig. 5). Rounded particles had relatively higher abundances at low wind speeds (<17 m s$^{-1}$), while core-shell particles increased in relative contribution with increasing wind speeds. Similar collections of SSA produced during CHAOS had identical relationships between relative

contributions of core-shell and rounded particles with wind speed as the ice crystal residuals, suggesting the INPs are subsets of all the observed SSA particle morphologies (Madawala et al., 2024, in review). The SSA particles collected were also analyzed for elemental composition by scanning electron microscopy coupled with energy dispersive X-ray spectroscopy (SEM-EDX) as





in Ault et al. (2013) and functional group characterization using atomic force microscopy-Photothermal Infrared spectroscopy (AFM-PTIR) following Or et al. (2018). SEM-EDX indicated rounded SSA particles were organic carbon throughout, while

core-shell particles had a cubic NaCl core and organic shell. Rounded particles had more diverse organic functional groups (fatty acids, complex sugars and in some cases traces of sulfates and carbonates), and their composition was similar at both 9.57 and 18.72 m s$^{-1}$. Core-shell particles were dominated by aliphatic compounds at 9.57 m s$^{-1}$, with the addition of oxygenated organics at 18.72 m s$^{-1}$ (Madawala et al., 2024, in review).

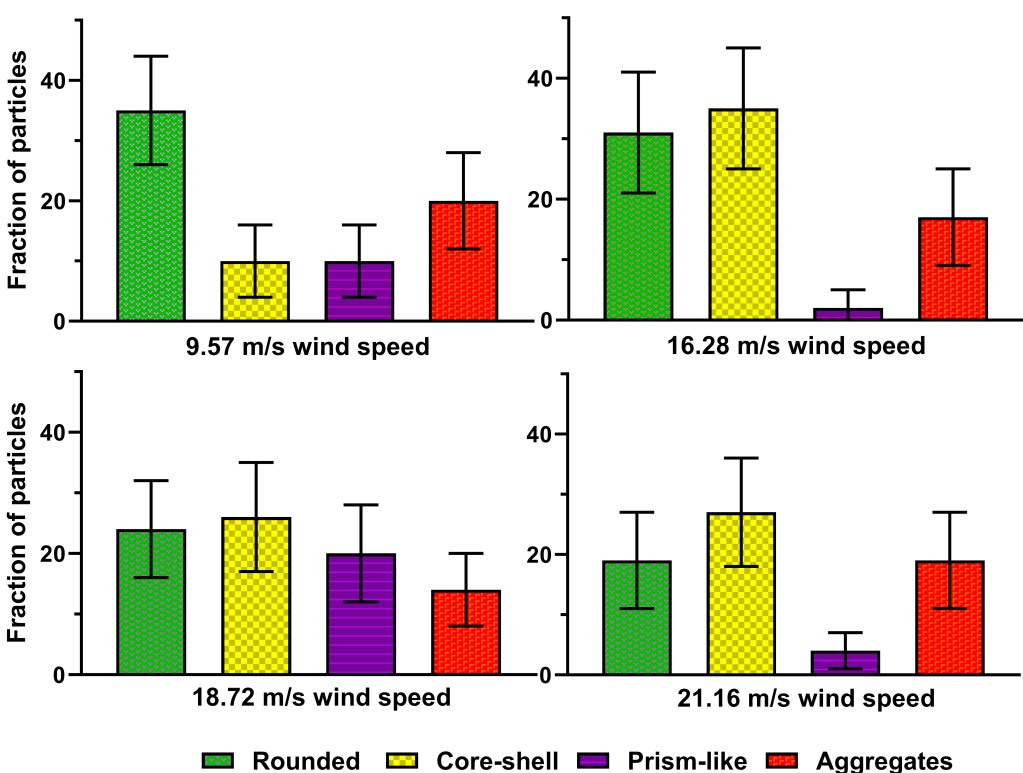

**Figure 5.** Percentage of particles from each morphological category observed during CHAOS at four of the measured wind speeds (9.57, 16.28, 18.72, and 21.16 m s$^{-1}$). For each sample, the individual particles (N = 50 for each sample) characterized were in the volume-equivalent diameter range of 0.05 – 1.0 μm.

Viscoelastic response distance (VRD), which is related to the viscosity of the material (Lee et al., 2020; Kaluarachchi et al., 2022a), as well as particle phase state (Lee et al., 2017, 2020) was quantified for core-shell ice crystal residuals at 20 % and 60 % RH (Table A1). As anticipated due to the hygroscopicity of SSA, the fraction of semisolid shells increased between 20



and 60 % RH at all wind speeds. Below 17 m s$^{-1}$, the shell region of core-shell particles was predominantly solid at 20 % RH, while at higher wind speeds, shells were more often semisolid even at low RH. VRD measurements were only possible on semisolid shells, and were similar at both 20 % and 60 % RH for a given wind speed, but were higher for wind speeds >17 m s$^{-1}$. The increased abundance of semisolid shells with higher VRD is consistent with lower viscosity and the presence of more oxygenated chemical species in the shell region of core shell particles at higher wind speeds.

## 4 Conclusions

Initial results from the CHAOS campaign were presented here, which focused on the role of wind speed in the production of SSA and INPs, using the new SOARS wind-wave channel at the Scripps Institution of Oceanography. As expected from numerous field and laboratory measurements, SSA concentrations increased with increasing wind speed (Fig. 1). Enhanced particle concentrations were observed relative to Southern Ocean MBL measurements in a similar wind speed range (Moore et al., 2022) by maximum factors of ∼50, ∼45, and ∼7 for particle number, surface area, and volume, respectively. INP concentrations were broadly consistent with previous measurements from the Southern Ocean (McCluskey et al., 2018a; Schmale et al., 2019; Moore et al., 2024) and North Atlantic (McCluskey et al., 2018c), although SOARS concentrations were biased high and normalized concentrations biased low relative to ambient results (Fig. 2). This is likely related to the low sampling height over the water surface during CHAOS (0.6 m), which may capture more large particles than are typically sampled during ship-board or coastal campaigns where aerosol inlets may be 20+ m above sea level and/or offset from the shore. As a result, measurements from CHAOS likely represent interfacial values and may not be directly comparable to MBL or cloud-base measurements.

INP concentrations also generally increased with wind speed, especially for the CFDC measurements, as was observed in the Southern Ocean MBL (Moore et al., 2024). However, normalized INP concentrations decreased with increasing wind speeds during CHAOS, while the opposite relationship was observed in Moore et al. (2024) for the Southern Ocean (Fig. 2, Fig. 3). In addition to the low sampling inlet height and consequently lower particle losses, the fixed 1.3 amplitude scaling for wave height used during CHAOS may help explain this discrepancy. Further measurements where wind speed and wave amplitude are both varied to produce whitecap fractions representative of open ocean conditions (Monahan and Muircheartaigh, 1980) are required to separate these mechanisms. Additionally, the large spread and highly variable particle concentrations observed for both SSA and INPs during CHAOS complicated analysis and should be addressed through detailed estimates of particle losses within the SOARS channel and inlets and more systematic sampling of wind speeds than was possible during CHAOS due to time constraints. Seawater INE concentrations during CHAOS were stable and consistent with previous measurements at the SIO pier and in other regions (McCluskey et al., 2018a; Gong et al., 2020; Hartmann et al., 2021; Beall et al., 2022), indicating changes in atmospheric INPs during CHAOS were driven by wind speed and wave-breaking mechanics rather than variations in seawater chemistry or biology (Fig. A7).

Heat-stable organic INPs were the dominant composition at wind speeds below 15 m s$^{-1}$ (Fig. 4, Fig. A8), which corresponds to the DOC-type marine INP described in McCluskey et al. (2018b). A number of field measurements have also identified




similar small, heat-stable marine INPs (Rosinski et al., 1987; Wilson et al., 2015; Ladino et al., 2016; Alpert et al., 2022), although a second category of larger and protinaceous (heat-labile) marine INPs has also been observed in both field and laboratory measurements (Knopf et al., 2011; Wang et al., 2015; Irish et al., 2017; McCluskey et al., 2018b). At high wind speeds, peroxide-treated filter samples almost uniformly had higher INP concentrations than untreated samples (Fig. 4, Fig. A9), which has not been previously seen for marine INPs. We hypothesize that spume droplet production at higher wind speeds, coupled with the low height of the SOARS sampling inlet, may have allowed for the sampling of larger, aggregate particles containing multiple INPs, which were broken up through peroxide digestion. The composition of INPs emitted in such gels is unknown, since results from CHAOS are consistent with dust or other inorganic contaminants that are unaffected by peroxide digestion, or heat stable organics which are only released from the larger particle and not broken down due to the 20-min digestions performed here. The very short atmospheric lifetime of large (>10 μm) spume droplets may explain why this has not been seen in ambient measurements or laboratory experiments without wind (Wang et al., 2015; McCluskey et al., 2018b). Entrapment of INEs in gels may also play a role in their low number concentrations in seawater (McCluskey et al., 2018a; Gong et al., 2020; Hartmann et al., 2021; Beall et al., 2022) compared to terrestrial sources such as soil or mineral dust, fungi, and permafrost (O'Sullivan et al., 2014; Fröhlich-Nowoisky et al., 2015; Hill et al., 2016; Kanji et al., 2017; Barry et al., 2023) due to both reduced emissions of large particles and enhanced oceanic deposition through marine snow formation or other processes.

AFM 3D height images of collected ice crystal residuals were used to identify 6 dominant particle morphologies, which were similar to residual classifications during SeaSCAPE (DeMott et al., 2023). Rounded particles were the most abundant morphology at wind speeds <17 m s$^{-1}$, and core-shell particles dominated at higher wind speeds (Fig. 5). The abundance of core-shell particles with semisolid shells increased with wind speed, while the viscosity of the shells simultaneously decreased. This is consistent with an increasing contribution of oxygenated chemical species in the shells, which was also noted as an outcome of heterogeneous aging of INPs during SeaSCAPE (DeMott et al., 2023). It is possible the decreased viscosity and more complex chemical composition at high wind speeds is related to the enhancement in INP concentration following peroxide digestions through increased water solubility of the shells, as was observed during SeaSCAPE for aged SSA (Kaluarachchi et al., 2022a).

The CHAOS campaign represents a first attempt at using the new SOARS wind-wave channel to isolate individual factors impacting SSA and INP emissions from seawater. Additional experiments with co-varying wind speed and wave amplitude are ongoing, focusing initially on measuring SSA (and not INP) concentrations. This is intended to generate realistic whitecap fraction-wind speed pairings to increase comparability with ambient data. Both SSA and INP concentrations measured by the CFDC increased with wind speed during CHAOS, as expected. IS measurements of INP concentration demonstrated a less clear trend with wind speed, which may be due to the use of separate inlets with different particle losses. The very low sampling height during CHAOS (0.6 m) relative to ambient (several to 20+ m) may have led to decreased losses of large particles, and requires further study before the comparability of such interfacial measurements to ambient marine boundary layer observations can be assessed. A mechanism involving spume droplet production of aggregate particles was proposed to explain the unexpected results of peroxide digestions of IS filters collected at high wind speeds, which also requires further



observations to evaluate. Following additional characterization of particle losses in SOARS and aerosol sampling inlets, and utilizing measurements with paired wind speed and whitecap fraction, future studies in the SOARS facility will be well poised to answer remaining questions about SSA and INP emissions as a function of wind speed, wave state, and temperature.

*Data availability.* Data presented in this study is in the process of being archived in the Dryad repository (https://datadryad.org/stash) and
545 will be available online soon.

**Appendix A**

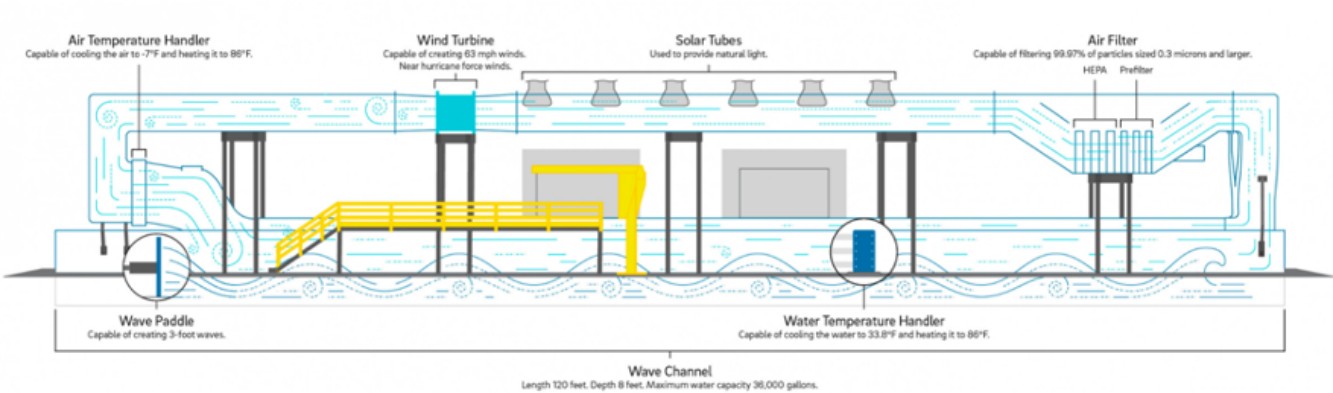

**Figure A1.** Schematic of the Scripps Ocean-Atmosphere Research Simulator (SOARS) wind-wave channel at the Scripps Institution of Oceanography showing key features relevant for SSA production and seawater biology.

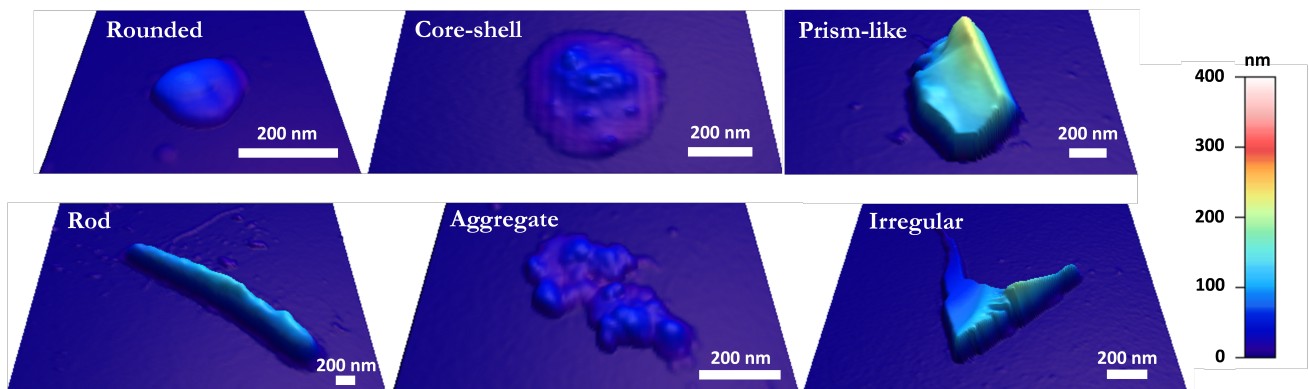

**Figure A2.** Selected illustrative AFM 3D-height images of six main particle morphological categories (rounded, core-shell, prism-like, rod, aggregate and irregular) identified at four wind speeds of 9.57, 16.28, 18.72, and 21.16 m s$^{-1}$. Images were all collected at 20 % RH.



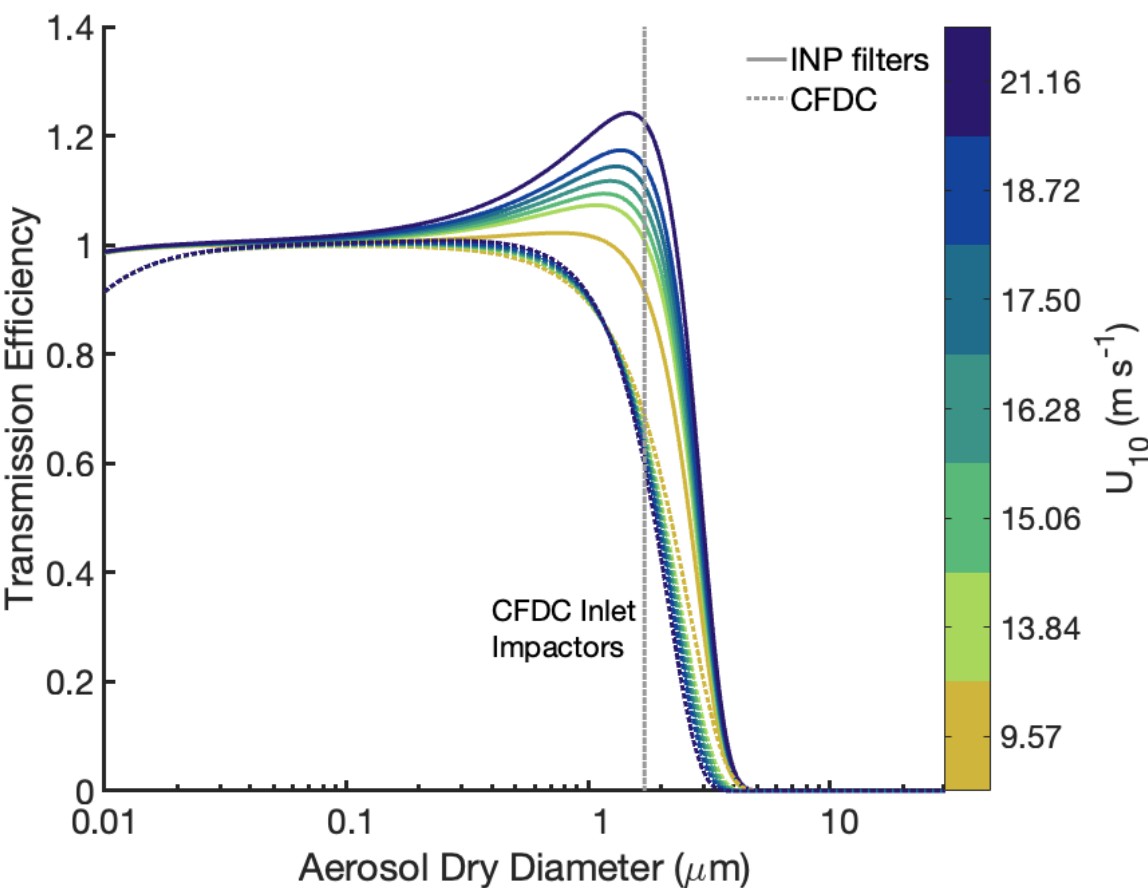

**Figure A3.** Estimated particle transmission efficiency during CHAOS for particles reaching either the CFDC (SEMS + APS) or INP filters (SMPS + APS), based on the different inlet geometries. These theoretical calculations used the von der Weiden et al. (2009) Particle Loss Calculator. Calculations were performed for the whole inlet in aerodynamic diameter, with a particle density $\rho$=1 and later corrected for expected particle density, water uptake, and shape factor (Sec. 2.3). Colors indicate the wind speed of the measurement, with INP filter curves in solid lines and CFDC curves in dashed lines.



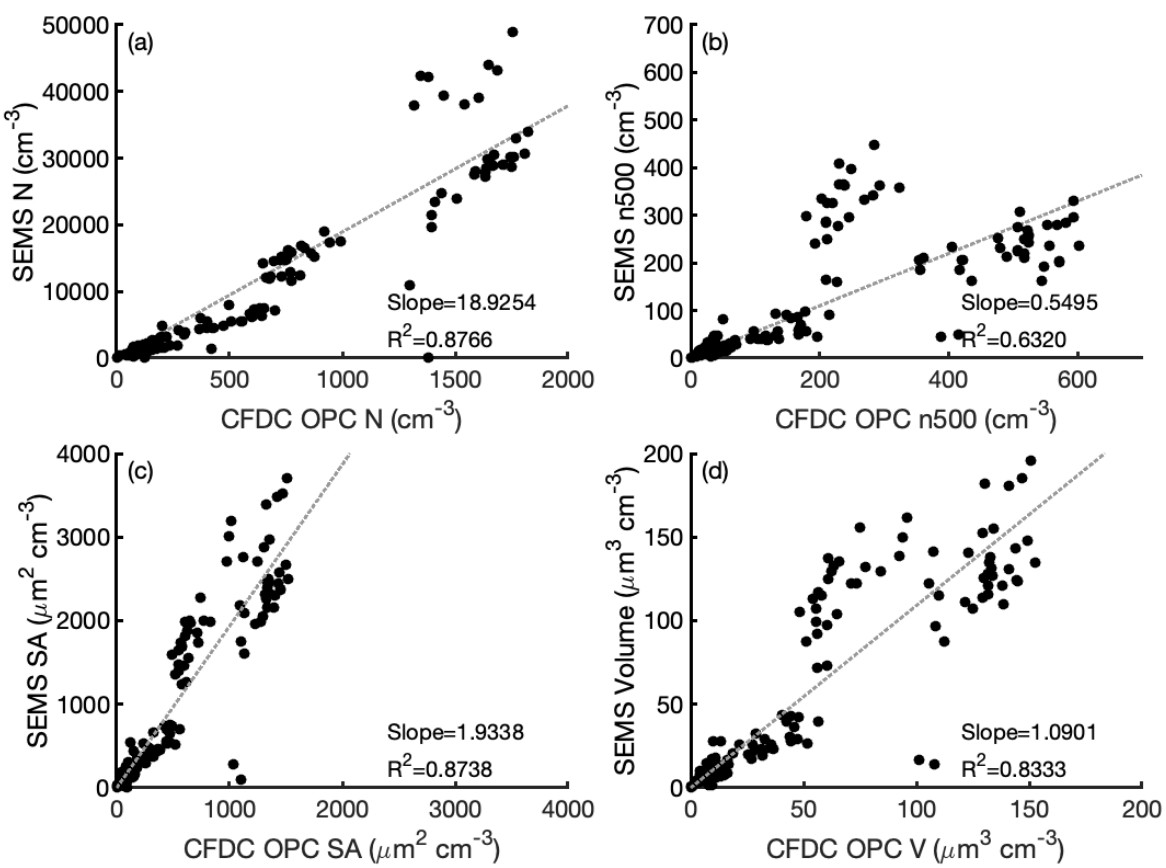

**Figure A4.** Correction factors derived for the CFDC OPC based on SEMS + APS aerosol measurements, for total particle number (a), n500 (b), particle surface area (c) and particle volume (d) concentrations.

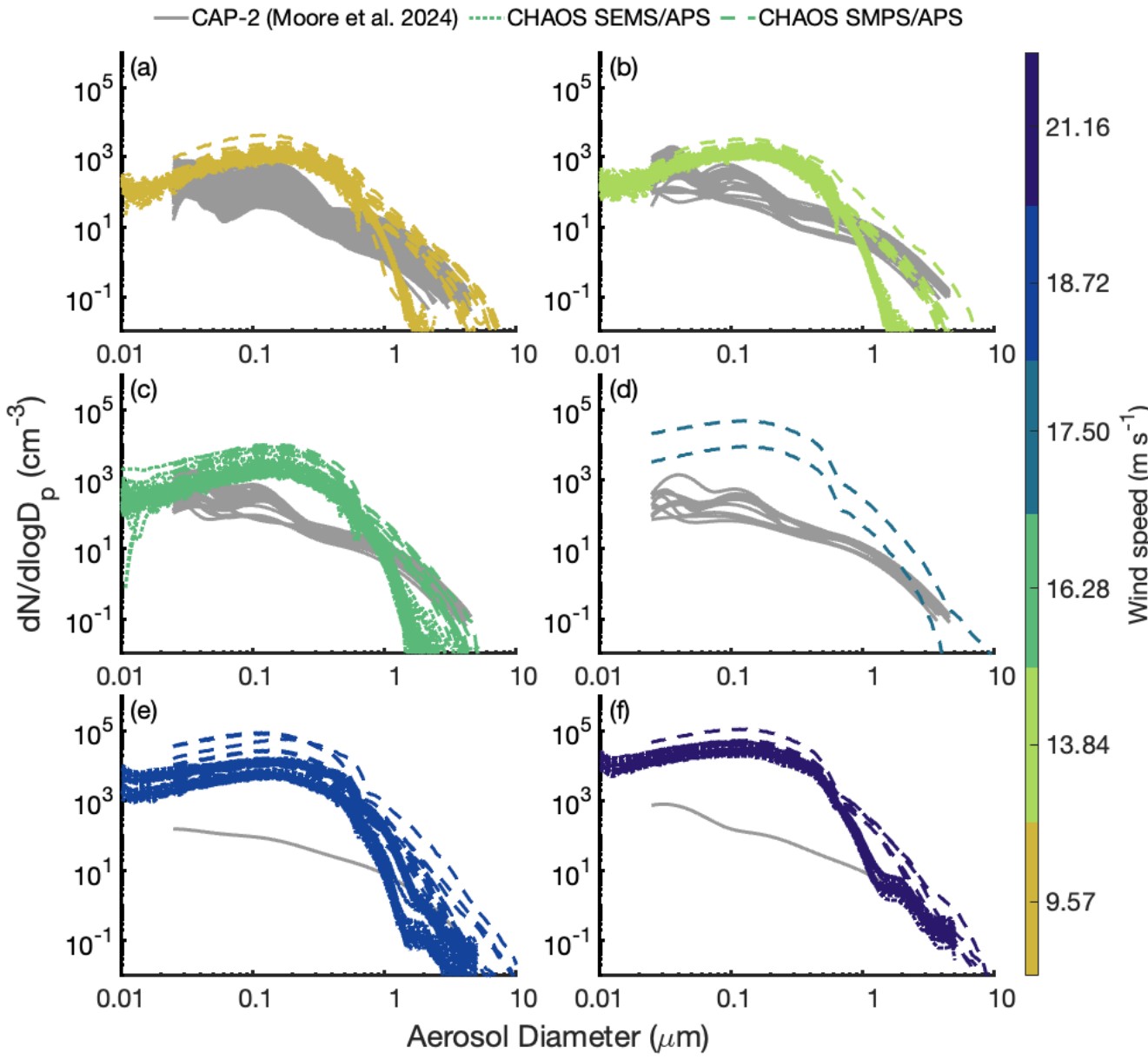

**Figure A5.** CHAOS aerosol size distributions at (a) 9.57 m s$^{-1}$ (yellow), (b) 13.84 m s$^{-1}$ (light green), (c) 16.28 m s$^{-1}$ (green), (d) 17.50 m s$^{-1}$ (light blue), (e) 18.72 m s$^{-1}$ (dark blue), and (f) 21.16 m s$^{-1}$ (dark purple). Measurements from the SEMS + APS are shown in the colored dotted lines, SMPS + APS observations in the colored dashed lines, and observations from CAP-2 (Moore et al., 2024), in the solid gray lines, if available. Measurements from CAP-2 are shown if they are within ±0.5 m s$^{-1}$ of the SOARS U$_{10}$ values.







**Figure A6.** Time series of CFDC INP (a) number concentration, (b) normalized by n500 ($N_{n500}$), (c) normalized by aerosol surface area ($N_s$) and (d) normalized by aerosol volume ($N_v$) during CHAOS. Observations are colored by the wind speed during each measurement period.



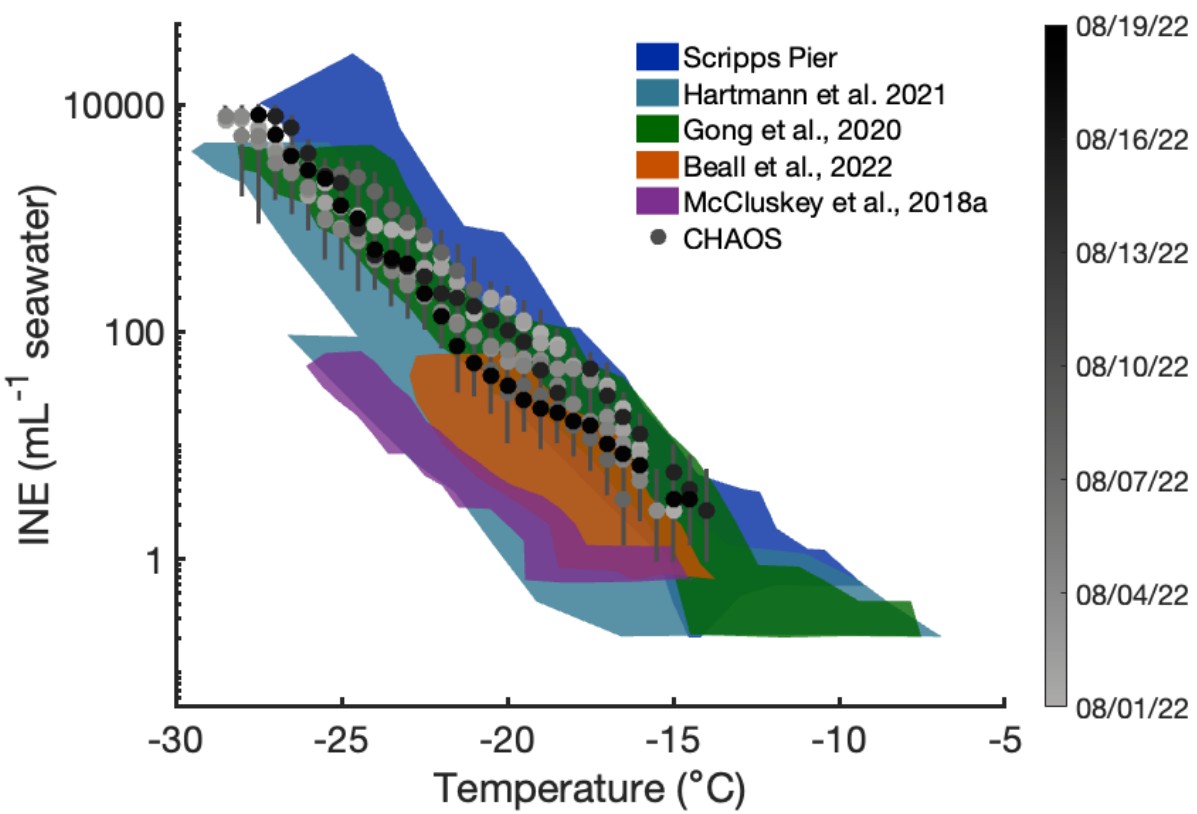

**Figure A7.** Seawater INE temperature spectra during CHAOS (grey circles), colored by collection date. Colored patches indicate comparisons with measurements from the Scripps Pier (dark blue), Barents Sea (Hartmann et al., 2021, light blue), mid-Atlantic (Gong et al., 2020, green), North Indian Ocean (Beall et al., 2022, orange), and Southern Ocean (McCluskey et al., 2018a, purple).





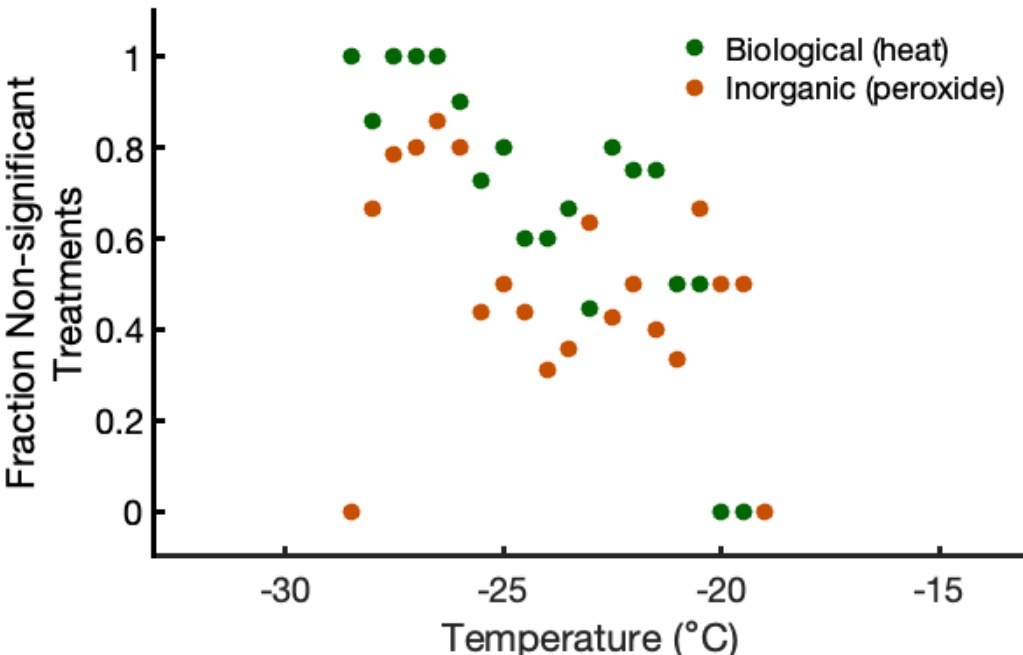

**Figure A8.** Fraction of INP filter treatment results that are not statistically different from the base spectra at the 95% confidence level. Results are shown as a function of temperature, with results for biological INPs (heat treatment) in green, and inorganic INPs (peroxide treatment) in orange.



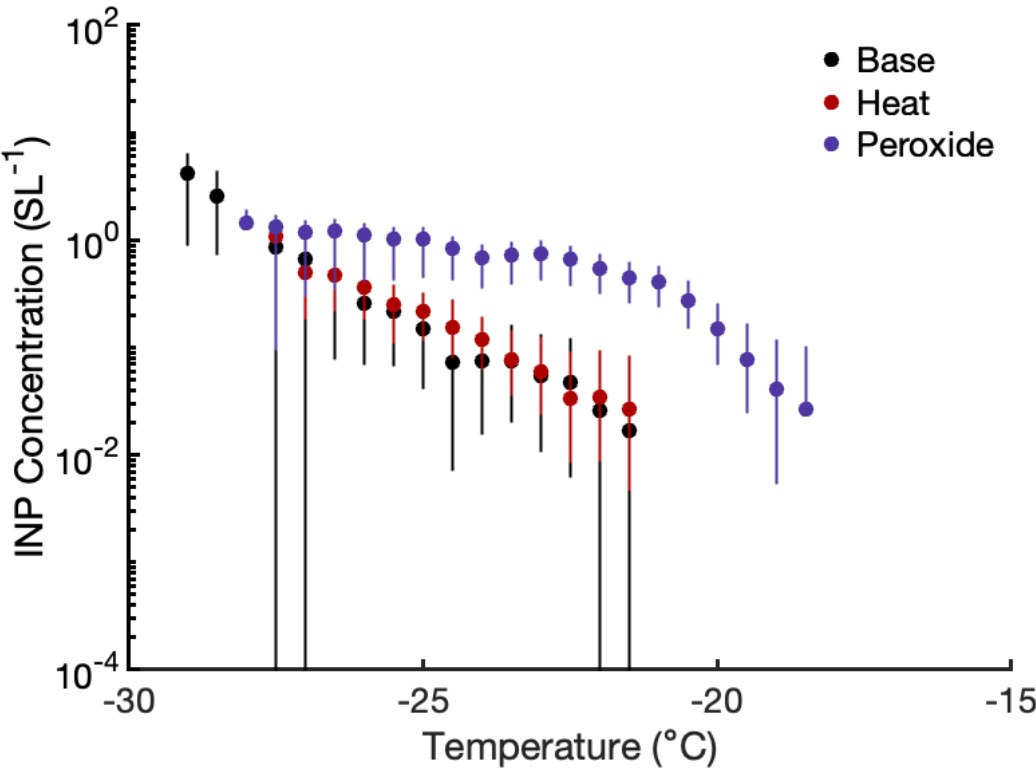

**Figure A9.** An example IS filter temperature spectra from 8/5/22 (18.72 m s$^{-1}$), with base measurements in black, heat-treated in red, and peroxide-treated in purple.



**Table A1.** Summary of the average (± one standard deviation) percentage of core-shell INPs with solid, semisolid, and liquid shells emitted at different wind speeds (9.57, 16.28, 18.72, and 21.16 m s⁻¹), as well as the average and range of viscoelastic response distances (VRD) measured for particles with semisolid shells. Measurements were made at 20 % and 60 % RH.

| Wind speed (m s$^{-1}$) | RH ( %) | Solid ( %) | Semisolid ( %) | Liquid ( %) | VRD* (nm) | VRD range* (nm) |
|---|---|---|---|---|---|---|
| 9.57 | 20 | 60 ± 24 | 0 | 40 ± 26 | N/A | N/A |
|  | 60 | 0 | 60 ± 26 | 40 ± 27 | 0.6 ± 0.1 | 0.5 – 0.7 |
| 16.28 | 20 | 95 ± 1 | 5 ± 1 | 0 | 0.6 ± 0.0 | 0.6 |
|  | 60 | 47 ± 16 | 53 ± 16 | 0 | 0.8 ± 0.4 | 0.5 – 1.5 |
| 18.72 | 20 | 50 ± 24 | 42 ± 23 | 8 ± 1 | 2.7 ± 1.9 | 0.7 – 4.4 |
|  | 60 | 14 ± 1 | 71 ± 22 | 14 ± 1 | 2.8 ± 2.3 | 0.8 – 5.4 |
| 21.16 | 20 | 46 ± 18 | 46 ± 18 | 8 ± 1 | 1.5 ± 1.2 | 0.5 – 3.6 |
|  | 60 | 0 | 71 ± 21 | 29 ± 10 | 1.8 ± 1.6 | 0.5 – 3.8 |

*Data reported only for core-shell particles with organic coatings classified as semisolid

*Author contributions.* MDS, GBD, CL, and KAP designed the overall CHAOS campaign. KAM and PJD led the collection and processing of online INP measurements; KAM and TCJH led the collection and analysis of offline INP measurements, with assistance from SG. CKM and AVT led analyses and interpretation of AFM measurements of collected INPs. RJLIII and CDC led the collection and analysis of
550 supplementary aerosol measurements. KAM and CKM created figures. KAM led the writing and editing of this article, with contributions from all the other authors.

*Competing interests.* There are no competing interests to declare.

*Acknowledgements.* This work was funded by the National Science Foundation (NSF) through the NSF Center for Aerosol Impacts on Chemistry of the Environment (CAICE) under award CHE-1801971. The authors thank the entire CHAOS team for their hard work through-
555 out the campaign. Particular thanks are due to Joseph Mayer, Robert Klidy, and the team at the Scripps Institution of Oceanography Marine Science Development Center for their engineering support. KAM acknowledges support by an NSF Graduate Research Fellowship under Grant 006784. Any opinions, findings and conclusions or recommendations expressed in this material are those of the authors and do not necessarily reflect the views of the National Science Foundation.



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
