# Peer review of "Wind-driven Emission of Marine Ice Nucleating Particles in the Scripps Ocean-Atmosphere Research Simulator (SOARS)"

_EGUsphere, 2024_

## Referee Comment (RC1)

**Summary**

Whilst the authors state that these are "initial results" from the CHAOS study using the Scripps SOARS facility, this is somewhat of an understatement. The results are very comprehensive and will significantly contribute to knowledge of SSA and INP emission processes associated with wind driven mechanisms over marine sources. Based on the current results and the comprehensive and open approach, there is promise of much more to come and will encourage better and more careful approaches to ambient oceanic measurements of INP.

Despite a rapid expansion over the past decade in INP observations, using well-developed and characterised techniques such as the CSU Ice Spectrometer, there has been less progress in accurately quantifying INP emission fluxes from different source and how these emission mechanisms respond to a wide range of meteorological drivers. Whilst INP are now regarded as critical to fully understanding and predicting aerosol-cloud feedback processes, particularly in marine environments, discriminating them by their chemical composition and morphological characteristics in field studies within timescales relevant to these emission processes for more accurate parameterisation in current climate models is still limited and significantly more work is required here. As the authors state, their results reproduce well numerous previous laboratory and ambient observation studies in the Southern Ocean wrt ambient SSA concentrations due mainly to wind speed variation. However, this study has also emphasised the importance of inclusion of more detailed INP morphological analytics, particularly surface and volume site density (Niemand et al. 2012 and citations within), for future ambient studies, which is an additional excellent step forward and which more of the INP community should adopt.

The introduction-review section is extremely comprehensive and very useful.  [21,22,32]  the authors do reference some observations in the Arctic [e.g. 21, 31, 32] as a suggestion, with respect to seasonal ambient measurements of INP and INP contributions due to biological the authors could refer to the long-term INP monitoring study in the Arctic by Freitas, G.P. et al., (2023) "Regionally sourced bioaerosols drive high-temperature ice nucleating particles in the Arctic", Nat. Commun., 14, 5997, https://doi.org/10.1038/s41467-023-41696-7, 2023. Which similar heat labile INP measurement techniques combined with UVLIF single particle measurement techniques (MBS).

It would be interesting for future SOARS studies to look to include single particle fluorescence spectral signature measurements. This would potentially expand the comparison of future SOARS studies with rapidly growing aerosol databases using integrated optoelectronic UVLIF and particle morphology measurements to discriminate INP by bioaerosol and non-bioaerosol classes and their sources, as in the Freitas (2023) study.

The authors also very carefully and clearly identify potential limitations wrt comparison of laboratory studies with ambient ship-borne observations of INP concentrations due to aerosol inlet sampling heights (0.5 to 20+ m) and coastal offset distance for tower measurements as well as the ongoing problem of inlet loss issues (well-known and difficult practical issues). These can significantly bias results as also clearly demonstrated here (and which the authors highlight in their conclusions) making it difficult to intercompare oceanic surface layer interfacial process generated concentrations with relevant cloud base level concentrations due to ship deployment differences in sampling approaches. This can be a particular problem for coarse mode particles, particularly for INP concentration quantification which can vary very significantly depending on the measurement technique employed and so needs to be better addressed in future studies.

The sections on methodologies and measurement techniques are very comprehensive, particularly regarding the INP and chemical composition- and the AFM/3D imaging and force spectroscopy techniques and phase states. This provides confidence in the results. The overall description demonstrates an excellent integrated measurement technique approach with careful consideration of the uncertainties, which has been a limitation in some previous studies.

One common limitation in the study the authors highlight is the total number of individual particles that can be studied with traditional AFM, which the authors rely on for aspects of their analysis. To address this they employ a standard probability distribution analysis to assess statistical significance of their results via probability distribution curves using the standard Markov chain Monte Carlo method e.g. for morphological

classes. Whilst this is acceptable to enable the conclusions presented here perhaps in future the authors should investigate more up to date neural net supervised and unsupervised approaches which are now being applied routinely to airborne single particle measurements generated by widely used biogenic aerosol integrated-optoelectronic-holographic spectrometers to identify specific emission mechanisms including particle breakup. This could allow for easier intercomparison of the SOARS data with ambient real-time ambient measurements in the future to better identify and quantify those emission mechanisms that dominate in ambient environments.

It would be interesting and very useful for future SOARS studies to include/provide single particle UVLIF+ fluorescence spectra to add to the potential to ibtercompare their results with growing observational databases using this approach to segregate aerosol type. Perhaps therefore the authors could briefly mention the work by Freitas et al. "Regionally sourced bioaerosols drive high-temperature ice nucleating particles in the Arctic", Nat. Commun., 14, 5997, https://doi.org/10.1038/s41467-023-41696-7, 2023, (differentiation of sources as mentioned is this and similar studies can still be a significant uncertainty here due to meteorological driver initiating changes in emission vs particle lifetime) which compares INP measurement with biogenic discrimination and comparison with real-time UVLIF measurements at a remote Arctic station (albeit at a high elevation). I believe this would contribute to adding to the contextualisation of the SOARS data here and may be helpful to encourage community engagement to better integrate future databases generated by laboratory, in situ integrated ship and aircraft campaigns as well as long-term monitoring stations.

A relevant citation the authors should perhaps also include and comment on is, Freitas et al. (2022), Emission of primary bioaerosol particles from Baltic seawater, https://doi.org/10.1039/d2ea00047d, Environmental Science: Atmospheres, Volume 2, Issue 5, 2022. Their conclusions from a ship-borne and spray chamber study were limited but suggested very low bioaerosol contributions (< 0.5%). The importance of this contribution needs to be assessed and I wonder if here the concentration detection accuracy of biogenic INP can be quantified/highlighted a little more from this study?

An interesting conclusion from this study, is that "seawater ice nucleating entity concentrations during CHAOS were stable over time, indicating changes in atmospheric INPs were driven by wind speed and wave-breaking mechanics rather than variations in seawater chemistry or biology." Can it be better contextualised that this is specific to the Pier sample site and perhaps state here how its variability compares to other locations with respect to changes in marine surface chemistry?

I found the section describing higher INP concentrations seen at high wind speeds interesting and the point regarding peroxide-treated filter samples generating uniformly higher INP concentrations than untreated samples even more interesting with respect to stable organic components of oceanic emissions which has consequences for future measurement technique assessment in these environments.

**Discussion**

Purely for discussion and not required for inclusion in this work, I wonder if these results may eventually be used to potentially link to more fundamental oceanic biogeochemistry cycles? Specifically with respect to marine-reduced organic nitrogen components of the oceanic N cycle. It has recently been observed e.g. that biologically rich oceanic environments appear exhibit much larger concentrations of gas phase urea over the lower MBL than previously thought and this may be responsible for enhanced redistribution of reduced N over seawater surfaces, although the impact on new particle formation and INP in marine environments is still being investigated. [Matthews, Emily, Bannan, Thomas J., Khan, M. Anwar H. et al. (20 more authors) (2023) *Airborne observations over the North Atlantic Ocean reveal the importance of gas-phase urea in the atmosphere.* Proceedings of the National Academy of Sciences of the United States of America. e2218127120. ISSN 1091-6490]

510 A9), which has not been previously seen for marine INPs. We hypothesize that spume droplet production at higher wind speeds, coupled with the low height of the SOARS sampling inlet, may have allowed for the sampling of larger, aggregate particles containing multiple INPs, which were broken up through peroxide digestion. The composition of INPs emitted in such gels is unknown, since results from CHAOS are consistent with dust or other inorganic contaminants that are unaffected by peroxide digestion, or heat stable organics which are only released from the larger particle and not broken down due to the 20-min

Overall, I found this study was excellently performed and the review of current knowledge in the field extremely useful. Their use of integrated INP and chemical composition/morphology measurements was excellent and serves as a useful reference for future studies. The interpretations and conclusions were also presented in an open manner with useful aims and objectives outlined for the community. I found this paper to be excellent therefore, bordering on exceptional (? although these are "preliminary results"), and easily worthy of publication in ACP. This paper will significantly contribute to enhancing scientific knowledge in this complex subject area and encouraging future, better integrated research in this field. I look forward to seeing more results from CHAOS/SOARS.

**Additional References**

Pereira Freitas, G., Kopec, B., Adachi, K., Krejci, R., Heslin-Rees, D., Yttri, K. E., Hubbard, A., Welker, J. M., and Zieger, P.: Contribution of fluorescent primary biological aerosol particles to low-level Arctic cloud residuals, Atmos. Chem. Phys., 24, 5479–5494, https://doi.org/10.5194/acp-24-5479-2024, 2024.

Pereira Freitas, G., Adachi, K., Conen, F., Heslin-Rees, D., Krejci, R., Tobo, Y., Yttri, K. E., and Zieger, P.: Regionally sourced bioaerosols drive high-temperature ice nucleating particles in the Arctic, Nat. Commun., 14, 5997, https://doi.org/10.1038/s41467-023-41696-7, 2023.

---

## Author Comment (AC1)

Author Response to RC1 for ACP manuscript EGUSPHERE-2024-2159 (Moore et al.)

Our responses to each comment are provided in blue text, with the reviewer comments in black. Line and figure numbers refer to the line numbers in the original submission, for consistency with the Reviewer comments. Quotations in red were added to the revised manuscript.
* * *
**Summary**

Whilst the authors state that these are "initial results" from the CHAOS study using the Scripps SOARS facility, this is somewhat of an understatement. The results are very comprehensive and will significantly contribute to knowledge of SSA and INP emission processes associated with wind driven mechanisms over marine sources. Based on the current results and the comprehensive and open approach, there is promise of much more to come and will encourage better and more careful approaches to ambient oceanic measurements of INP.

Despite a rapid expansion over the past decade in INP observations, using well-developed and characterised techniques such as the CSU Ice Spectrometer, there has been less progress in accurately quantifying INP emission fluxes from different source and how these emission mechanisms respond to a wide range of meteorological drivers. Whilst INP are now regarded as critical to fully understanding and predicting aerosol-cloud feedback processes, particularly in marine environments, discriminating them by their chemical composition and morphological characteristics in field studies within timescales relevant to these emission processes for more accurate parameterisation in current climate models is still limited and significantly more work is required here. As the authors state, their results reproduce well numerous previous laboratory and ambient observation studies in the Southern Ocean wrt ambient SSA concentrations due mainly to wind speed variation. However, this study has also emphasised the importance of inclusion of more detailed INP morphological analytics, particularly surface and volume site density (Niemand et al. 2012 and citations within), for future ambient studies, which is an additional excellent step forward and which more of the INP community should adopt.

**Response:** We thank the reviewer for their careful reading of the manuscript and the thoughtful comments and discussion which have improved it.

The introduction-review section is extremely comprehensive and very useful. [21,22,32] the authors do reference some observations in the Arctic [e.g. 21, 31, 32] as a suggestion, with respect to seasonal ambient measurements of INP and INP contributions due to biological the authors could refer to the long-term INP monitoring study in the Arctic by Freitas, G.P. et al., (2023) "Regionally sourced bioaerosols drive high temperature ice nucleating particles in the Arctic", Nat. Commun., 14, 5997, https://doi.org/10.1038/s41467-023-41696-7, 2023. Which similar heat labile INP measurement techniques combined with UVLIF single particle measurement techniques (MBS).

**Response:** Thank you for pointing out this very interesting paper, which discusses much-needed seasonal/annual INP and PBAP measurements in the Arctic. Freitas et al. 2023 concludes that terrestrially-sourced PBAP were the source of the high temperature INPs observed during the summer at Zeppelin Observatory (Svalbard). Our article focuses on the role of marine-derived

INPs, so we do not feel this reference belongs in the introduction, which includes citations of studies that identified marine-derived INPs in the Arctic.

It would be interesting for future SOARS studies to look to include single particle fluorescence spectral signature measurements. This would potentially expand the comparison of future SOARS studies with rapidly growing aerosol databases using integrated optoelectronic UVLIF and particle morphology measurements to discriminate INP by bioaerosol and non-bioaerosol classes and their sources, as in the Freitas (2023) study.

Response: A similar instrument to the MBS used in the Freitas et al. 2023 study (WIBS NEO) was used during the CHAOS campaign, although due to technical difficulties very little data was collected. During follow-up experiments in the SOARS channel, a WIBS NEO and other bioaerosol measurements (including DNA sequencing) were performed, and the results will be presented in an upcoming manuscript.

The authors also very carefully and clearly identify potential limitations wrt comparison of laboratory studies with ambient ship-borne observations of INP concentrations due to aerosol inlet sampling heights (0.5 to 20+ m) and coastal offset distance for tower measurements as well as the ongoing problem of inlet loss issues (well-known and difficult practical issues). These can significantly bias results as also clearly demonstrated here (and which the authors highlight in their conclusions) making it difficult to intercompare oceanic surface layer interfacial process generated concentrations with relevant cloud base level concentrations due to ship deployment differences in sampling approaches. This can be a particular problem for coarse mode particles, particularly for INP concentration quantification which can vary very significantly depending on the measurement technique employed and so needs to be better addressed in future studies.

Response: We agree wholeheartedly, and hope this can be addressed in future studies, particularly during the upcoming International Polar Year (2032-3033).

The sections on methodologies and measurement techniques are very comprehensive, particularly regarding the INP and chemical composition- and the AFM/3D imaging and force spectroscopy techniques and phase states. This provides confidence in the results. The overall description demonstrates an excellent integrated measurement technique approach with careful consideration of the uncertainties, which has been a limitation in some previous studies.

One common limitation in the study the authors highlight is the total number of individual particles that can be studied with traditional AFM, which the authors rely on for aspects of their analysis. To address this they employ a standard probability distribution analysis to assess statistical significance of their results via probability distribution curves using the standard Markov chain Monte Carlo method e.g. for morphological classes. Whilst this is acceptable to enable the conclusions presented here perhaps in future the authors should investigate more up to date neural net supervised and unsupervised approaches which are now being applied routinely to airborne single particle measurements generated by widely used biogenic aerosol integrated-optoelectronic-holographic spectrometers to identify specific emission mechanisms including particle breakup. This could allow for easier intercomparison of the SOARS data with ambient

real-time ambient measurements in the future to better identify and quantify those emission mechanisms that dominate in ambient environments.

**Response:** We appreciate the suggestion and will look into these neural net approaches for future studies.

It would be interesting and very useful for future SOARS studies to include/provide single particle UVLIF+ fluorescence spectra to add to the potential to ibtercompare their results with growing observational databases using this approach to segregate aerosol type. Perhaps therefore the authors could briefly mention the work by Freitas et al. "Regionally sourced bioaerosols drive high-temperature ice nucleating particles in the Arctic", Nat. Commun., 14, 5997, https://doi.org/10.1038/s41467-023-41696-7, 2023, (differentiation of sources as mentioned is this and similar studies can still be a significant uncertainty here due to meteorological driver initiating changes in emission vs particle lifetime) which compares INP measurement with biogenic discrimination and comparison with real-time UVLIF measurements at a remote Arctic station (albeit at a high elevation). I believe this would contribute to adding to the contextualisation of the SOARS data here and may be helpful to encourage community engagement to better integrate future databases generated by laboratory, in situ integrated ship and aircraft campaigns as well as long-term monitoring stations.

A relevant citation the authors should perhaps also include and comment on is, Freitas et al. (2022), Emission of primary bioaerosol particles from Baltic seawater, https://doi.org/10.1039/d2ea00047d, Environmental Science: Atmospheres, Volume 2, Issue 5, 2022. Their conclusions from a ship-borne and spray chamber study were limited but suggested very low bioaerosol contributions (< 0.5%). The importance of this contribution needs to be assessed and I wonder if here the concentration detection accuracy of biogenic INP can be quantified/highlighted a little more from this study?

**Response:** As mentioned above, follow-on experiments have and are being conducted in the SOARS channel, using some of the lessons learned from CHAOS. One set of experiments included measurements from a different UVLIF instrument to the MBS used in Freitas et al. (2023), the WIBS NEO, as well as other bioaerosol measurements (ie DNA sequencing), and those results will be reported in an upcoming publication. However, in the meantime, we have also added an additional paragraph to Sec. 3.2 (following line 424), which discusses the INP composition results from CHAOS, and compares them to both Freitas et al. (2022) and (2023). The new text is copied below:

"Concentrations of heat-labile INPs during CHAOS ranged from $3.1 \times 10^{-3}$ to $4.3 \times 10^{-2}$ L$^{-1}$, and when normalized by aerosol n500, from $4.0 \times 10^{-8}$ to $1.2 \times 10^{-6}$. Heat treatments which produced increased INP concentrations over the untreated filters are excluded from these ranges, since they are not representative of the emission of biological INPs during CHAOS, but instead of post-emission modification. Samples meeting this criteria all had estimated biological INP fractions of 1, were at relatively warm temperatures ($\geq$ -24 °C), and were predominantly collected at 9.6 m s$^{-1}$ wind speed, in accordance with Fig. 4a-c. Both the concentrations and high biogenic fraction of these warm-temperature INPs from CHAOS are in agreement with recent INP measurements in the Arctic (Hartmann et al., 2020; Freitas et al., 2023), although Hartmann et al. (2020)

concluded marine INPs were the likely source, while Freitas et al. (2023) determined local terrestrial primary biological aerosol particles (PBAPs) were the dominant contributor to their measurements. Using a plunging jet chamber to produce SSA, Freitas et al. (2022) estimated the production of PBAPs from Baltic seawater to be ~1 in every $10^4$ particles larger than 0.8 μm. This is about 3 orders of magnitude larger than the median proportion of biological INPs to total particles larger than 0.5 μm during CHAOS (~6 in every $10^7$), indicating that while marine biogenic particles can act as INPs, only a small fraction are able to do so, at least for temperatures ≥ -24 °C."

An interesting conclusion from this study, is that "seawater ice nucleating entity concentrations during CHAOS were stable over time, indicating changes in atmospheric INPs were driven by wind speed and wavebreaking mechanics rather than variations in seawater chemistry or biology." Can it be better contextualized that this is specigic to the Pier sample site and perhaps state here how its variability compares to other locations with respect to changes in marine surface chemistry?

**Response:** We have added additional details to Sec. 3.1 (following line 406) which provides additional context for the CHAOS measurements and reiterates the need for future experiments with a variety of seawater biological and chemical conditions. The added text is copied below:

"Seawater biology and chemistry, as well as air and water temperature, were not controlled during CHAOS and were allowed to vary throughout the experiments. This resulted in variations in seawater chlorophyll $a$, total organic carbon (TOC), temperature, salinity, and nutrient concentrations, among other factors (Fig. A2). As a result of collecting seawater from the SIO pier to fill the SOARS channel, the CHAOS measurements may be more representative of mid-latitude coastal marine regions than remote or polar ocean environments. In addition, the seawater was relatively warm (~25 °C) as well as high in silicates, so additional measurements under a range of biogeochemical conditions are needed to assess the robustness of these findings."

I found the section describing higher INP concentrations seen at high wind speeds interesting and the point regarding peroxide-treated filter samples generating uniformly higher INP concentrations than untreated samples even more interesting with respect to stable organic components of oceanic emissions which has consequences for future measurement technique assessment in these environments.

**Response:** We also found this to be a very interesting result and hope it will receive further study.

**Discussion**
Purely for discussion and not required for inclusion in this work, I wonder if these results may eventually be used to potentially link to more fundamental oceanic biogeochemistry cycles? Specifically with respect to marine-reduced organic nitrogen components of the oceanic N cycle. It has recently been observed e.g. that biologically rich oceanic environments appear exhibit much larger concentrations of gas phase urea over the lower MBL than previously thought and

this may be responsible for enhanced redistribution of reduced N over seawater surfaces, although the impact on new particle formation and INP in marine environments is still being investigated. [Matthews, Emily, Bannan, Thomas J., Khan, M. Anwar H. et al. (20 more authors) (2023) Airborne observations over the North Atlantic Ocean reveal the importance of gas phase urea in the atmosphere. Proceedings of the National Academy of Sciences of the United States of America. e2218127120. ISSN 1091-6490]

**Response:** This is a very interesting comment, although outside the scope of this current study. During and after CHAOS, the SOARS channel has been used for measurements of air-water partitioning of gases and future plans include studying atmospheric aging of particles and new particle formation. A large oxidation chamber is attached to the SOARS channel and can help answer some of these questions once testing is completed. Additionally, studies which will include more focus on and control of seawater chemistry and biology, and the links between seawater biogeochemistry and oceanic emissions of gases and particles are planned for the future.

510 A9), which has not been previously seen for marine INPs. We hypothesize that spume droplet production at higher wind speeds, coupled with the low height of the SOARS sampling inlet, may have allowed for the sampling of larger, aggregate particles containing multiple INPs, which were broken up through peroxide digestion. The composition of INPs emitted in such gels is unknown, since results from CHAOS are consistent with dust or other inorganic contaminants that are unaffected by peroxide digestion, or heat stable organics which are only released from the larger particle and not broken down due to the 20-min

**Response:** This appears to be copied from the submitted manuscript, lines 510-515.

Overall, I found this study was excellently performed and the review of current knowledge in the field extremely useful. Their use of integrated INP and chemical composition/morphology measurements was excellent and serves as a useful reference for future studies. The interpretations and conclusions were also presented in an open manner with useful aims and objectives outlined for the community. I found this paper to be excellent therefore, bordering on exceptional (? although these are "preliminary results"), and easily worthy of publication in ACP. This paper will significantly contribute to enhancing scientific knowledge in this complex subject area and encouraging future, better integrated research in this field. I look forward to seeing more results from CHAOS/SOARS.

**Response:** Thank you, we greatly appreciate the comments and feedback.

**Additional References**
Pereira Freitas, G., Kopec, B., Adachi, K., Krejci, R., Heslin-Rees, D., Yttri, K. E., Hubbard, A., Welker, J. M., and Zieger, P.: Contribution of fluorescent primary biological aerosol particles to low-level Arctic cloud residuals, Atmos. Chem. Phys., 24, 5479–5494, https://doi.org/10.5194/acp-24-5479-2024, 2024.

Pereira Freitas, G., Adachi, K., Conen, F., Heslin-Rees, D., Krejci, R., Tobo, Y., Yttri, K. E., and Zieger, P.: Regionally sourced bioaerosols drive high-temperature ice nucleating particles in the Arctic, Nat. Commun., 14, 5997, https://doi.org/10.1038/s41467-023-41696-7, 2023.

---

## Author Comment (AC2)

Author Response to RC3 for ACP manuscript EGUSPHERE-2024-2159 (Moore et al.)

Our responses to each comment are provided in blue text, with the reviewer comments in black. Line and figure numbers refer to the line numbers in the original submission, for consistency with the Reviewer comments. Quotations in red were added to the revised manuscript.
* * *
**Review of "Wind-driven Emission of Marine Ice Nucleating Particles in the Scripps Ocean-Atmosphere Research Simulator (SOARS)" by Moore et al.**

This work addresses the transfer of INPs from water to air via wind induced wave breaking processes.

The authors present an impressive new facility, the Scripps Ocean-Atmosphere Research Simulator (SOARS) wind-wave channel and describe results from a mesocosm campaign named CHAOS (CHaracterizing Atmosphere-Ocean parameters in SOARS). The authors have studied the effect of windspeed on emission of sea spray aerosol and properties of ice nucleating particles from seawater. A wide range of advanced instrumentation was applied to analyze sea spray aerosol and INPs.

Overall, the manuscript presents an amazing new facility and a large body of data, information and thoughts which can serve as inspiration for other studies.

I have some suggestions for improving the manuscript. After consideration of the comments/suggestions below, I recommend publication and look forward to following future findings from the new SOARS facility.

**Response:** We thank the reviewer for their detailed reading of the manuscript and for helpful suggestions that have improved it.

**Main comments**

As I understand, the results presented in the manuscript are based on two fillings of the channel with seawater each lasting several days. I miss one or more overview figures showing how the experiments evolved with time and showing how basic parameters such as temperature of water and air, as well as windspeed varied with time from start to end in each of the two experiments. It could also be marked when sampling of different types were performed. Such figures could be provided as supporting material. This will help the reader get a better basis for the discussion.

**Response:** We have added an overview figure to Appendix A (Fig. A2 in the revised manuscript) which shows details of the two water fills described in this manuscript. Parameters such as wind speed, seawater chlorophyll $a$ and total organic carbon (TOC), air and water temperature, water salinity, and selected seawater nutrient concentrations are shown. In addition, the timing of IS filter collections for INPs and details on lighting and water fill timing are indicated in this new figure. It is copied below:

[Figure]

**Figure A2.** Overview of SOARS parameters during CHAOS, including (a) wind speed and IS INP filter sampling times, (b) chlorophyll *a* and total organic carbon (TOC) concentrations, (c) air and water temperature and seawater salinity, and (d) select seawater nutrient concentrations. Nutrient concentrations shown in (d) are: nitrite ($NO_2^-$), phosphate ($PO_4^{3-}$), ammonium ($NH_4^+$), silicates, and nitrate ($NO_3^-$). The switch between the third (August 1-12) and fourth (August 14-26) fills of the SOARS channel is indicated in all panels by the dashed gray line. The period when the PAR LEDs were utilized in addition to the solar tubes for lighting is indicated by the yellow bar at the top of panel (b).

Abstract:

Regarding the sentence: "unlike recent measurements from the SOurther Ocean, real-time and offline INP observations during CHAOS exhibited opposite relationships with wind speed which may be related to sampling inlet differences"

I think the abstract would be more informative if the authors explained the relationship from this study explicitly and then say it is opposite to previous field work.

**Response:** The sentence quoted here was intended to indicate that the online (CFDC) and offline (IS filter) measurements from CHAOS exhibited opposite relationships with wind speed compared to each other. The CFDC results from CHAOS agree with recent field measurements. We have clarified this by rewording this sentence as follows: "In agreement with recent Southern Ocean measurements, online INP concentrations during CHAOS showed an increasing relationship with wind speed, whereas offline CHAOS INP concentrations did not, which may be related to sampling inlet differences."

Methods:

The level of detail given varies, but I assume it is to focus on what is important for the current data-set – for example, it is stated that wind turbine setpoints are calibrated using a TSI Inc. model 9545-A air velocity meter while no details are given on the camera used to infer whitecap coverage. If this manuscript will serve as a reference for future studies in the facility it might be useful if such details are provided. I also realize that wind speed is in focus in this work and that other details may be given in future publications which the authors already mention are in progress or planning, and so perhaps that is why they are omitted?

**Response:** A detailed overview paper describing the capabilities of the SOARS channel is in progress and will be submitted soon. Because that paper is not yet available to reference, this manuscript provides a broad overview of the facility and more detailed information about the CHAOS campaign and the most relevant components of the SOARS facility, but does not describe every component or capability in great detail.

Were the solar simulators turned on during the current study? What were the brand and wavelength spectrum of the solar simulators and the PAR LEDs?

**Response:** The solar simulators were on throughout August 2022, and the PAR LEDs were on during August 2-12. The solar simulators are manufactured by SolaTube and match the wavelength spectrum of ambient light, since they simply redirect ambient light into the SOARS channel. The PAR LEDs (ONCE AgriShift MLS) produce broad spectrum white light (400-700 nm). These details have been added to the methods section 2.1.

What were the temperatures of the seawater and the ambient air during the two periods, did they vary?

**Response:** As mentioned in Section 2.1, water and air temperature were not controlled during CHAOS and were allowed to vary according to the ambient temperature. The new Figure A2 includes a time series of both seawater and air temperatures.

This is a huge facility and water volume, how is the channel cleaned?

**Response:** Between different wind speed measurements, the air ducts are rinsed with freshwater and then high wind speeds (21 m s$^{-1}$) are used to remove particle build up from the channel and duct walls. Finally, air is recycled through the HEPA and Carbon MERV filters until particle concentrations have reached baseline levels prior to ramping the wind speed up to the new value.

Between water fills, the water in the channel is drained, and the entire SOARS channel is pressure washed with freshwater and then manually scrubbed. Following another freshwater rinse, the channel is ready to be refilled. These details have been added to the methods section 2.1.

Iine 160: It is a bit unclear what/when the sampling periods took place and how long they were. Sampling periods for different types of sampling (e.g. aerosol filters, INP) could be indicated in overview figures as suggested above.

**Response:** Sampling periods varied in both time and length each day depending on a variety of factors, including technical difficulties with the new paddle assembly, but were generally several hours per wind speed, with multiple sampling periods per day. Wind speeds and INP filter collections are now indicated in the new summary Fig. A2.

Line 275: why is a density of 2 assumed?. I think there is a typo in the unit: should be  g cm-3 not g cm-1?

**Response:** The density is based on Zieger et al. (2017), and a citation has been added to the text. Thank you for catching that typo, the unit is indeed g cm$^{-3}$, and this has been corrected.

Figure A3: how should a transmission efficiency larger than one be understood? I suggest to also show the corresponding figure using the corrected values of density etc.

**Response:** A transmission efficiency larger than one indicates particles of that size are more concentrated in the air stream than their ambient concentration (in the SOARS channel). This enhancement is due to the relative air speeds in the channel and inlet, as well as the inlet orientation and geometry. Sorry for the confusion, but Fig. A3 (now Fig. A4) does in fact show the calculated transmission efficiencies after adjusting for expected particle density, water uptake, and shape factor. We have replaced "later" with "then" in the figure caption and text to hopefully clarify this.

Figure A4:  The description of normalization and figure A4 would benefit from further explanation.  Under which conditions are the data taken – is the slope the correction factor? What are the uncertainties? Is n300 nm sometimes used for normalization – this is difficult to understand from the text.

**Response:** Correction factors for the CFDC OPC were derived because its lower size bound is ~300 nm, so it misses smaller particles. Particle concentrations > 300 nm were never used for normalization, on the contrary, they were corrected to better represent the ambient aerosol concentrations in the SOARS channel. Methods section 2.3 has been reworded to clarify this, and that the correction factors were derived from simultaneous SEMS + APS and CFDC OPC measurements (at the same wind speed). The caption for Fig. A4 (now Fig. A5) has been updated to clarify that the slopes shown in the figure are the correction factors applied to the CFDC OPC data. An additional supplementary table (Table C2) has been added, which lists the correction factors and uncertainties for each aerosol parameter. Table C2 is copied below:

**Table C2.** Correction factors (and 95% confidence bounds) for total particle number, number >500 nm diameter (n500), surface area, and volume concentrations measured by the CFDC OPC, which were derived by comparison with simultaneous (same wind speed) SEMS + APS measurements.

| Aerosol Parameter | Correction Factor | $R^2$ |
|---|---|---|
| Number ($cm^{-3}$) | $18.93 \pm 0.74$ | 0.88 |
| n500 ($cm^{-3}$) | $0.55 \pm 0.04$ | 0.63 |
| Surface Area ($\mu m^2\ cm^{-3}$) | $1.93 \pm 0.07$ | 0.87 |
| Volume ($\mu m^3\ cm^{-3}$) | $1.09 \pm 0.05$ | 0.83 |

In lines 150-153 white cap fraction during CHAOS is discussed, but no data or values are given, is it possible to give such numbers?

**Response:** White cap fraction in the SOARS channel will be discussed in detail in the SOARS methods paper, which is about to be submitted. Since the actual white cap fraction data is not a focus of this paper, we have added the estimated white cap fraction for 18.5 m s$^{-1}$ and 1.3 amplitude scaling ($6.44 \pm 1.53\%$) to the text, but will leave a comparison with open-ocean measurements and other wind speeds and amplitudes to the SOARS methods paper.

As I understand the wind speeds shown in all figures are the values extrapolated to 10 m height (U10) from the measured windspeed at 0.6 m in SOARS – is this correct? The extrapolation is based on Hsu et al. 1994 – I suggest providing the value of p used, also, I was wondering, is it necessary/justified to give the extrapolated U10 windspeed with two decimals (m/s) on basis of Hsu et al?

**Response:** This is exactly correct. We have added the value of *P* used (0.11) from Hsu et al. (1994) and also clarified that near-neutral stability was assumed. We agree that two decimals are probably not justified and have reduced wind speeds to one decimal in all text and figures.

**Results and discussion**

Figure 2A –To avoid misunderstandings, I suggest that the equations for activated fraction, surface site density and volume site density are provided. Is surface and volume densities also normalized only to particles larger than 500 nm? (perhaps it does not matter so much as the larger particles dominate these distributions?)

**Response:** The equations for INP activated fraction and surface and volume site density are now given in Appendix B and referenced in Sec. 3.1. Surface and volume site density are calculated from total particle surface area or volume; only N$_{n500}$ is limited to particles >500 nm.

I suggest showing the surface and volume distributions for the size distributions in Figure A5 in and additional figure.

**Response:** Since surface area and volume distributions are not discussed anywhere in the manuscript, we are not clear what value this would add for the reader. A study focused on SSA fluxes in SOARS is in progress, which will include the suggested distributions.

Regarding the INE measurements: it is concluded: "The INE stability across multiple fills of the SOARS channel and over time with the same water indicates the observed INP-wind speed relationships were driven by wind-wave interactions rather than biological activity" and in the abstract it says that sea water ice nucleating entity concentrations during CHAOS were stable over time and therefore changes in INPs were driven by windspeed and wavebreaking mechanisms and not seawater chemistry or biology. Were any data on seawater chemical composition or biology available to further support this interesting conclusion? This could be interesting to look further into in future studies.

**Response:** Yes, measurements of chlorophyll *a*, total organic carbon (TOC), microbial cell counts, and nutrient concentrations were made throughout CHAOS. A selection of these data are now shown in the new summary Fig. A2. Future studies in CHAOS will focus more on the relationship between seawater biology and chemistry and subsequent SSA and gas production.

**Minor comments**

What is the material of the paddle?

**Response:** The paddle is made of a fiberglass and foam core with an epoxy coating. It is strengthened with titanium rails, and the edges that contact the ceramic bearing pads on the walls and floor of the channel are covered in teflon slides. These details have been added to the methods Sec. 2.1.

The sentence about SMA (line 25) is in between two sentences about SSA – since SMA is not SSA this is a bit confusing. I suggest moving the SMA sentence to the end of the paragraph.

**Response:** We have rearranged the paragraph as suggested.

Figure A1: it would be nice with indication of length scales on the figure.

**Response:** Fig. A1 has been updated with larger labels and a length scale.

Line 116: The function of the MERV 8 and potassium permanganate filters should be explained.

**Response:** All the air filters installed in SOARS are intended to prevent ambient particle and VOC contamination from impacting measurements inside the SOARS headspace. This has been additionally clarified in Sec. 2.1.

Line 198: I find something is strange in the formulation of this sentence - how can filters be collected without airflow?

**Response:** Blank filters are "collected" without applying airflow to the filter in order to estimate the process blank of all steps included in INP filter collection. This includes installation of the filters in the filter housing, transport to the SOARS channel, and connection to the sampling tubing. Airflow is not applied because they are intended to estimate INP background contamination and not sample concentrations. As described later in Sec. 2.2.2, the blank filters were used to correct the sample filter INP concentrations to remove background INP concentrations.

Something is wrong with the URL to the Hsu paper in the reference list.

**Response:** Thank you, this has been corrected.

---

## Author Comment (AC3)

Author Response to RC2 for ACP manuscript EGUSPHERE-2024-2159 (Moore et al.)

Our responses to each comment are provided in blue text, with the reviewer comments in black. Line and figure numbers refer to the line numbers in the original submission, for consistency with the Reviewer comments. Quotations in red were added to the revised manuscript.
* * *
This work was well designed, the experiments and analysis were thoroughly carried out. I especially appreciate the detailed description of the experiments, which could be really helpful for other scientists in the community to conduct similar work. Overall, this study is excellent.

I have only one comment. The "less clear trend with wind speed" of INP concentration from IS measurements was attributed to "different averaging times, differences in inlet orientations or locations, or differences in the aerosol sampled". I have no objections about the possibility of these factors. However, there might be two other reasons, which could be somehow more important. First, for IS method the particles were formed from evaporation of seawater droplets, later collected on filters, and then washed off by water. The resulted solution should be very similar to the original seawater, except they might have different concentrations. It should have similar ice-nucleating properties as the seawater, which could be confirmed by the similar slopes of the INP spectra for filters in Figure 2a and those for seawater in Figure A7. Therefore, IS actually measured seawater-like solutions. As seawater used in experiments kept the same, it is understandable IS method did not see the trend as CFDC method. Second, the two methods measured INP for particles with different sizes. CFDC measured polydispersed particles. The different size distribution of particles generated by different wind speed would of course result in different INP concentrations. However, the IS equivalently measured several groups (meaning each dilution) of monodispersed seawater-droplets generated particles. For each group (one dilution), the droplets had similar size and might have similar INP properties. If you maintain the concentration of solution by using different amount of water to wash the filters according to the mass collected, you might get even less clear trend with wind speed for IS measurements.

**Response:** We thank the reviewer for reading the manuscript and providing thoughtful comments. Our apologies, but we do not follow the additional reasons provided for the discrepancy between the IS and CFDC relationships with wind speed. Both the IS and CFDC sampled the same aerosol at the same time, which was produced by wave breaking and bubble bursting within the SOARS channel. Both sampled polydisperse SSA, although differences in the inlets used by the CFDC and IS are hypothesized to be responsible for some of the differences in INP concentration seen between the two instruments, with inlet losses at high wind speed potentially responsible for the unclear relationship between the IS INP concentrations and wind speed. Good agreement between the IS and CFDC at the same temperature is expected and has been seen in laboratory and field measurements in multiple locations and with different aerosol types, including marine aerosol (e.g. DeMott et al., 2016). Some examples of detailed intercomparisons between different INP measurement methods, including both CFDC and IS, can be found in DeMott et al. (2017) and DeMott et al. (2018). One difference between the CFDC and IS measurements is that due to instrument constraints, the CFDC only measures INPs <2.4 μm (50% aerodynamic diameter cut size), whereas the IS is only limited by inlet transmission, which varies based on the specific set up used. Since the CFDC cannot measure the

largest INPs, it is possible, and has been observed in some cases, that the IS concentrations are higher than the CFDC, but they should still show a similar relationship with wind speed. Neither of these were observed during CHAOS.

The source of aerosol for both INP measurements during CHAOS was seawater, and so INPs sampled by both the IS and CFDC are expected to have similar ice-nucleation properties to each other and the source seawater. Despite the seawater source being the same when the wind speed was varied, the concentration of aerosols increased due to enhanced SSA production, and the CFDC also observed an increase in INP concentration with wind speed. The CFDC was operated under water-supersaturated conditions (RH>100%) during CHAOS, to measure INPs active in the immersion freezing mode. The IS, which requires re-suspension of particles collected on filters in water prior to measurement, also measures INPs active in the immersion freezing mode. Although the measurement technique of the IS is different from that of the CFDC, the drop-freezing method (Vali 1971) of the IS also measures the full INP population present, the same as the CFDC. Since they sampled the same polydisperse aerosol and both measured INPs active in the immersion freezing mode, the IS was also expected to observe an increase in INP concentration with wind speed, the same as the CFDC.

References:

DeMott, P. J., Hill, T. C. J., McCluskey, C. S., Prather, K. A., Collins, D. B., Sullivan, R. C., et al. (2016). Sea spray aerosol as a unique source of ice nucleating particles. *Proceedings of the National Academy of Sciences*, *113*(21), 5797–5803. https://doi.org/10.1073/pnas.1514034112

DeMott, P. J., Hill, T. C. J., Petters, M. D., Bertram, A. K., Tobo, Y., Mason, R. H., et al. (2017). Comparative measurements of ambient atmospheric concentrations of ice nucleating particles using multiple immersion freezing methods and a continuous flow diffusion chamber. *Atmospheric Chemistry and Physics*, *17*(18), 11227–11245. https://doi.org/10.5194/acp-17-11227-2017

DeMott, P. J., Möhler, O., Cziczo, D. J., Hiranuma, N., Petters, M. D., Petters, S. S., et al. (2018). The Fifth International Workshop on Ice Nucleation phase 2 (FIN-02): laboratory intercomparison of ice nucleation measurements. *Atmospheric Measurement Techniques*, *11*(11), 6231–6257. https://doi.org/10.5194/amt-11-6231-2018

Vali, G. (1971). Quantitative Evaluation of Experimental Results on the Heterogeneous Freezing Nucleation of Supercooled Liquids. *Journal of the Atmospheric Sciences*, *28*(3), 402–409. https://doi.org/10.1175/1520-0469(1971)028<0402:QEOERA>2.0.CO;2